# Mosaic: in-memory computing and routing for small-world spike-based neuromorphic systems

Thomas Dalgaty [1,3], Filippo Moro [1,3], Yiğit Demirağ [2,3], Alessio De Pra [1], Giacomo Indiveri [2], Elisa Vianello [1] & Melika Payvand [2] ✉

The brain's connectivity is locally dense and globally sparse, forming a small-world graph—a principle prevalent in the evolution of various species, suggesting a universal solution for efficient information routing. However, current artificial neural network circuit architectures do not fully embrace small-world neural network models. Here, we present the neuromorphic Mosaic: a non-von Neumann systolic architecture employing distributed memristors for in-memory computing and in-memory routing, efficiently implementing small-world graph topologies for Spiking Neural Networks (SNNs). We've designed, fabricated, and experimentally demonstrated the Mosaic's building blocks, using integrated memristors with 130 nm CMOS technology. We show that thanks to enforcing locality in the connectivity, routing efficiency of Mosaic is at least one order of magnitude higher than other SNN hardware platforms. This is while Mosaic achieves a competitive accuracy in a variety of edge benchmarks. Mosaic offers a scalable approach for edge systems based on distributed spike-based computing and in-memory routing.

Despite millions of years of evolution, the fundamental wiring principle of biological brains has been preserved: dense local and sparse global connectivity through synapses between neurons. This persistence indicates the efficiency of this solution in optimizing both computation and the utilization of the underlying neural substrate[1]. Studies have revealed that this connectivity pattern in neuronal networks increases the signal propagation speed[2], enhances echo-state properties[3] and allows for a more synchronized global network[4]. While densely connected neurons in the network are attributed to performing functions such as integration and feature extraction functions[5], long-range sparse connections may play a significant role in the hierarchical organization of such functions[6]. Such neural connectivity is called small-worldness in graph theory and is widely observed in the cortical connections of the human brain[2,7,8] (Fig. 1a, b). Small-world connectivity matrices, representing neuronal connections, display a distinctive pattern with a dense diagonal and progressively fewer connections between neurons as their distance from the diagonal increases (see Fig. 1c).

Crossbar arrays of non-volatile memory technologies e.g., Floating Gates[9], Resistive Random Access Memory (RRAM)[10–15], and Phase Change Memory (PCM)[16–19] have been previously proposed as a means for realizing artificial neural networks on hardware (Fig. 1d). These computing architectures perform in-memory vector-matrix multiplication, the core operation of artificial neural networks, reducing the data movement, and consequently the power consumption, relative to conventional von Neumann architectures[9,20–25].

However, existing crossbar array architectures are not inherently efficient for realizing small-world neural networks at all scales. Implementing networks with small-world connectivity in a large crossbar array would result in an under-utilization of the off-diagonal memory elements (i.e., a ratio of non-allocated to allocated connections > 10) (see Fig. 1d and Supplementary Note 1). Furthermore, the impact of analog-related hardware non-idealities such as current sneak-paths, parasitic resistance, and capacitance of the metal lines, as well as excessively large read currents and diminishing yield limit the maximum size of crossbar arrays in practice[26–28].

[1]CEA, LETI, Université Grenoble Alpes, Grenoble, France. [2]Institute of Neuroinformatics, University of Zurich and ETH Zurich, Zurich, Switzerland. [3]These authors contributed equally: Thomas Dalgaty, Filippo Moro, Yiğit Demirağ. ✉e-mail: melika@ini.uzh.ch

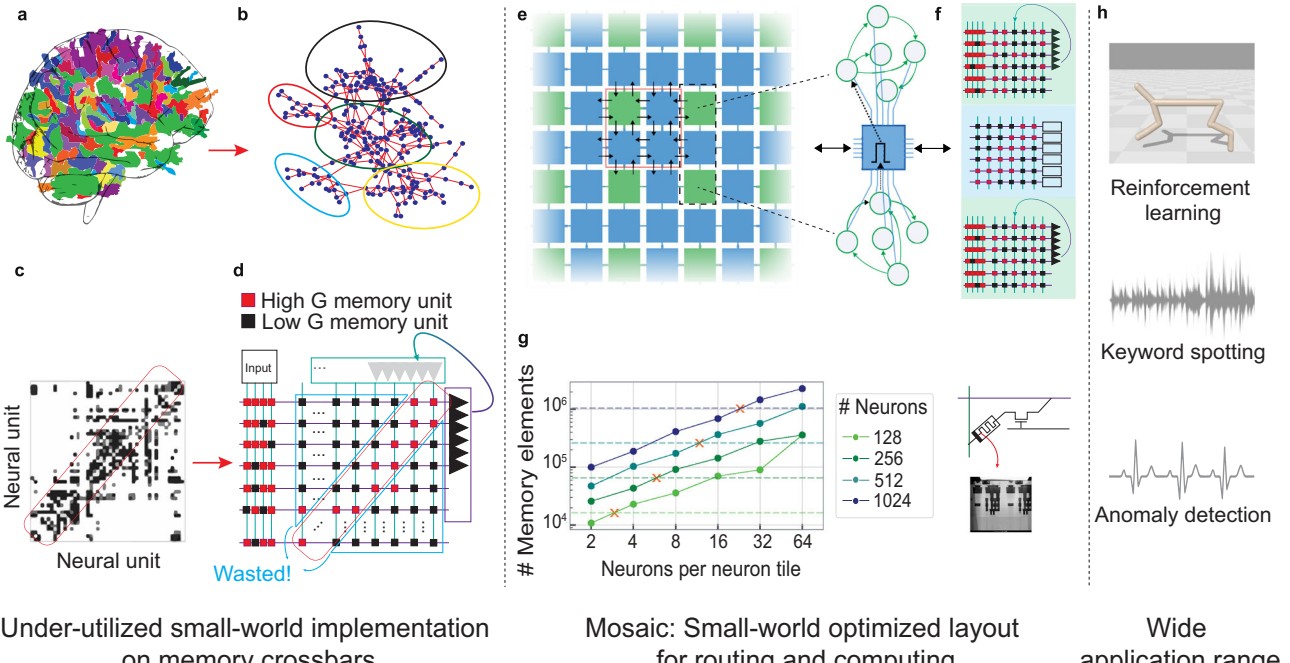

Under-utilized small-world implementation on memory crossbars

Mosaic: Small-world optimized layout for routing and computing

Wide application range

**Fig. 1 | Small-world graphs in biological network and how to build that into a hardware architecture for edge applications. a** Depiction of small-world property in the brain, with highly-clustered neighboring regions highlighted with the same color. **b** Network connectivity of the brain is a small-world graph, with highly clustered groups of neurons with sparse connectivity among them. **c** (adapted from Bullmore and Sporns[8]). The functional connectivity matrix which is derived from anatomical data with rows and columns representing neural units. The diagonal region of the matrix (darkest colors) contains the strongest connectivity which represents the connections between the neighboring regions. The off-diagonal elements are not connected. **d** Hardware implementation of the connectivity matrix of **c**, with neurons and synapses arranged in a crossbar architecture. The red squares represent the group of memory devices in the diagonal, connecting neighboring neurons. Black squares show the group of memory devices that are never programmed in a small-world network, and are thus highlighted as "wasted".

**e** The Mosaic architecture breaks the large crossbars into small densely-connected crossbars (green Neuron Tiles) and connects them through small routing crossbars (blue Routing Tiles). This gives rise to a distributed two-dimensional mesh, with highly connected clusters of neurons, connected to each other through routers. **f** The state of the resistive memory devices in Neuron Tiles determines how the information is processed, while the state of the routing devices determines how it is propagated in the mesh. The resistive memory devices are integrated into 130 nm technology. **g** Plot showing the required memory (number of memristors) as a function of the number of neurons per tile, for different total numbers of neurons in the network. The horizontal dashed line indicates the number of required memory bits using a fully-connected RRAM crossbar array. The cross (X) illustrates the cross-over point below which the Mosaic approach becomes favorable. **h** The Mosaic can be used for a variety of edge AI applications, benchmarked here on sensory processing and Reinforcement learning tasks.

These issues are also common to biological networks. As the resistance attenuates the spread of the action potential, cytoplasmic resistance sets an upper bound to the length of dendrites[1]. Hence, the intrinsic physical structure of the nervous systems necessitates the use of local over global connectivity.

Drawing inspiration from the biological solution for the same problem leads to (i) a similar optimal silicon layout, a small-world graph, and (ii) a similar information transfer mechanism through electrical pulses, or spikes. A large crossbar can be divided into an array of smaller, more locally connected crossbars. These correspond to the green squares of Fig. 1e. Each green crossbar hosts a cluster of spiking neurons with a high degree of local connectivity. To pass information among these clusters, small routers can be placed between them - the blue tiles in Fig. 1e. We call this two-dimensional systolic matrix of distributed crossbars, or tiles, the neuromorphic Mosaic architecture. Each green tile serves as an analog computing core, which sends out information in the form of spikes, while each blue tile serves as a routing core that spreads the spikes throughout the mesh to other green tiles. Thus, the Mosaic takes advantage of distributed and de-centralized computing and routing to enable not only in-memory computing, but also in-memory routing (Fig. 1f). Though the Mosaic architecture is independent of the choice of memory technology, here we are taking advantage of the resistive memory, for its non-volatility, small footprint, low access time and power, and fast programming[29].

Neighborhood-based computing with resistive memory has been previously explored through using Cellular Neural Networks[30,31], Self-organizing Maps (SOM)[32], and the cross-net architecture[33]. Though cellular architectures use local clustering, their lack of global connectivity limits both the speed of information propagation and their configurability. Therefore their application has been mostly limited to low-level image processing[34]. This also applies for SOMs, which exploit neighboring connectivity and are typically trained with unsupervised methods to visualize low-dimensional data[35]. Similarly, the crossnet architecture proposed to use distributed small tilted integrated crossbars on top of the Complementary Metal-Oxide-Semiconductor (CMOS) substrate, to create local connectivity domains for image processing[33]. The tilted crossbars allow the nano-wire feature size to be independent of the CMOS technology node[36]. However, this approach requires extra post-processing lithographic steps in the fabrication process, which has so far limited its realization.

Unlike most previous approaches, the Mosaic supports both dense local connectivity, and globally sparse long-range connections, by introducing re-configurable routing crossbars between the computing tiles. This allows to flexibly program specific small-world network configurations and to compile them onto the Mosaic for solving the desired task. Moreover, the Mosaic is fully compatible with standard integrated RRAM/CMOS processes available at the foundries, without the need for extra post-processing steps. Specifically, we have designed the Mosaic for small-world Spiking Neural Networks (SNNs), where the communication between the tiles is through electrical pulses, or spikes. In the realm of SNN hardware, the Mosaic goes beyond

the Address-Event Representation (AER)[37,38], the standard spike-based communication scheme, by removing the need to store each neuron's connectivity information in either bulky local or centralized memory units which draw static power and can consume a large chip area (Supplementary Note 2).

In this Article, we first present the Mosaic architecture. We report electrical circuit measurements from computational and Routing Tiles that we designed and fabricated in 130 nm CMOS technology co-integrated with Hafnium dioxide-based RRAM devices. Then, calibrated on these measurements, and using a novel method for training small-world neural networks that exploits the intrinsic layout of the Mosaic, we run system-level simulations applied to a variety of edge computing tasks (Fig. 1h). Finally, we compare our approach to other neuromorphic hardware platforms which highlights the significant reduction in spike-routing energy, between one and four orders of magnitude.

## Results

In the Mosaic (Fig. 1e), each of the tiles consists of a small memristor crossbar that can receive and transmit spikes to and from their neighboring tiles to the North (N), South (S), East (E) and West (W) directions (Supplementary Note 4). The memristive crossbar array in the green Neuron Tiles stores the synaptic weights of several Leaky Integrate and Fire (LIF) neurons. These neurons are implemented using analog circuits and are located at the termination of each row, emitting voltage spikes at their outputs[39]. The spikes from the Neuron Tile are copied in four directions of N, S, E and W. These spikes are communicated between Neuron Tiles through a mesh of blue Routing Tiles, whose crossbar array stores the connectivity pattern between Neuron Tiles. The Routing Tiles at different directions decides whether or not the received spike should be further communicated. Together, the two tiles give rise to a continuous mosaic of neuromorphic computation and memory for realizing small-world SNNs.

Small-world topology can be obtained by randomly programming memristors in a computer model of the Mosaic (see Methods and Supplementary Note 3). The resulting graph exhibits an intriguing set of connection patterns that reflect those found in many of the small-world graphical motifs observed in animal nervous systems. For example, central 'hub-like' neurons with connections to numerous nodes, reciprocal connections between pairs of nodes reminiscent of winner-take-all mechanisms, and some heavily connected local neural clusters[8]. If desired, these graph properties could be adapted on the fly by re-programming the RRAM states in the two tile types. For example, a set of desired small-world graph properties can be achieved by randomly programming the RRAM devices into their High-Conductive State (HCS) with a certain probability (Supplementary Note 3). Random programming can for example be achieved elegantly by simply modulating RRAM programming voltages[40].

For Mosaic-based small-world graphs, we estimate the required number of memory devices (synaptic weight and routing weight) as a function of the total number of neurons in a network, through a mathematical derivation (see Methods). Fig. 1g plots the memory footprint as a function of the number of neurons in each tile for different network sizes. Horizontal dashed lines show the number of memory elements using one large crossbar for each network size, as has previously been used for Recurrent Neural Networks (RNN) implementations[41]. The cross-over points, at which the Mosaic memory footprint becomes favorable, are denoted with a cross. While for smaller network sizes (i.e. 128 neurons) no memory reduction is observed compared to a single large array, the memory saving becomes increasingly important as the network is scaled. For example, given a network of 1024 neurons and 4 neurons per Neuron Tile, the Mosaic requires almost one order of magnitude fewer memory devices than a single crossbar to implement an equivalent network model.

## Hardware measurements

**Neuron tile circuits: small-worlds.** Each Neuron Tile in the Mosaic (Fig. 2a) is composed of multiple rows, a circuit that models a LIF neuron and its synapses. The details of one neuron row is shown in Fig. 2b. It has $N$ parallel one-transistor-one-resistor (1T1R) RRAM structures at its input. The synaptic weights of each neuron are stored in the conductance level of the RRAM devices in one row. On the arrival of any of the input events $V_{in<i>}$, the amplifier pins node $V_x$ to $V_{top}$, and thus a read voltage equivalent to $V_{top} - V_{bot}$ is applied across $G_i$, giving rise to current $i_{in}$ at $M_1$, and in turn to $i_{buff}$. This current pulse is mirrored through $I_w$ to the "synaptic dynamics" circuit, Differential Pair Integrator (DPI)[42], which low pass filters it through charging the capacitor $M_9$ in the presence of the pulse, and discharging it through current $I_{tau}$ in its absence. The charge/discharge of $M9$ generates an exponentially decaying current, $I_{syn}$, which is injected into the neuron's membrane potential node, $V_{mem}$, and charges capacitor $M_{13}$. The capacitor leaks through $M_{11}$, whose rate is controlled by $V_{lk}$ at its gate. As soon as the voltage developed on $V_{mem}$ passes the threshold of the following inverter stage, it generates a pulse, at $V_{out}$. The refractory period time constant depends on the capacitor $M_{16}$ and the bias on $V_{rp}$. (For a more detailed explanation of the circuit, please see Supplementary Note 5).

We have fabricated and measured the circuits of the Neuron Tile in a 130 nm CMOS technology integrated with RRAM devices[43]. The measurements were done on the wafer level, using a probe station shown in Fig. 2c. In the fabricated circuit, we statistically characterized the RRAMs through iterative programming[44] and controlling the programming current, resulting in nine stable conductance states, $G$, shown in Fig. 2d. After programming each device, we apply a pulse on $V_{in}<0>$ and measure the voltage on $V_{syn}$, which is the voltage developed on the $M_9$ capacitor. We repeat the experiment for four different conductance levels of 4 μS, 48 μS, 64 μS and 147 μS. The resulting $V_{syn}$ traces are plotted in Fig. 2e. $V_{syn}$ starts from the initial value close to the power supply, 1.2 V. The amount of discharge depends on the $I_w$ current which is a linear function of the conductance value of the RRAM, $G$. The higher the $G$, the higher the $I_w$, and higher the decrease in $V_{syn}$, resulting in higher $I_{syn}$ which is integrated by the neuron membrane $V_{mem}$. The peak value of the membrane potential in response to a pulse is measured across one array of 5 neurons, each with a different conductance level (Fig. 2g). Each pulse increases the membrane potential according to the corresponding conductance level, and once it hits a threshold, it generates an output spike (Fig. 2f). The peak value of the neuron's membrane potential and thus its firing rate is proportional to the conductance $G$, as shown in Fig. 2h. The error bars on the plot show the variability of the devices in the 4 kb array. It is worth noting that this implementation does not take into account the negative weight, as the focus of the design has been on the concept. Negative weights could be implemented using a differential signaling approach, by using two RRAMs per synapse[45].

**Routing tile circuits: connecting small-worlds.** A Routing Tile circuit is shown in Fig. 3a. It acts as a flexible means of configuring how spikes emitted from Neuron Tiles propagate locally between small-worlds. Thus, the routed message entails a spike, which is either blocked by the router, if the corresponding RRAM is in its High-Resistive State (HRS), or is passed otherwise. The functional principles of the Routing Tile circuits are similar to the Neuron Tiles. The principal difference is the replacement of the synapse and neuron circuit models with a simple current comparator circuit (highlighted with a black box in Fig. 3b). The measurements were done on the wafer level, using a probe station shown in Fig. 3c. On the arrival of a spike on an input port of the Routing Tile, $V_{in}<i>$, $0 < i < N$, a current proportional to $G_i$ flows to the device, giving rise to read current $i_{buff}$. A current comparator compares $i_{buff}$ against $i_{ref}$, which is a bias generated on chip by providing a voltage from the I/O pad to the gate of a transistor (not shown in Fig. 3). The $I_{ref}$

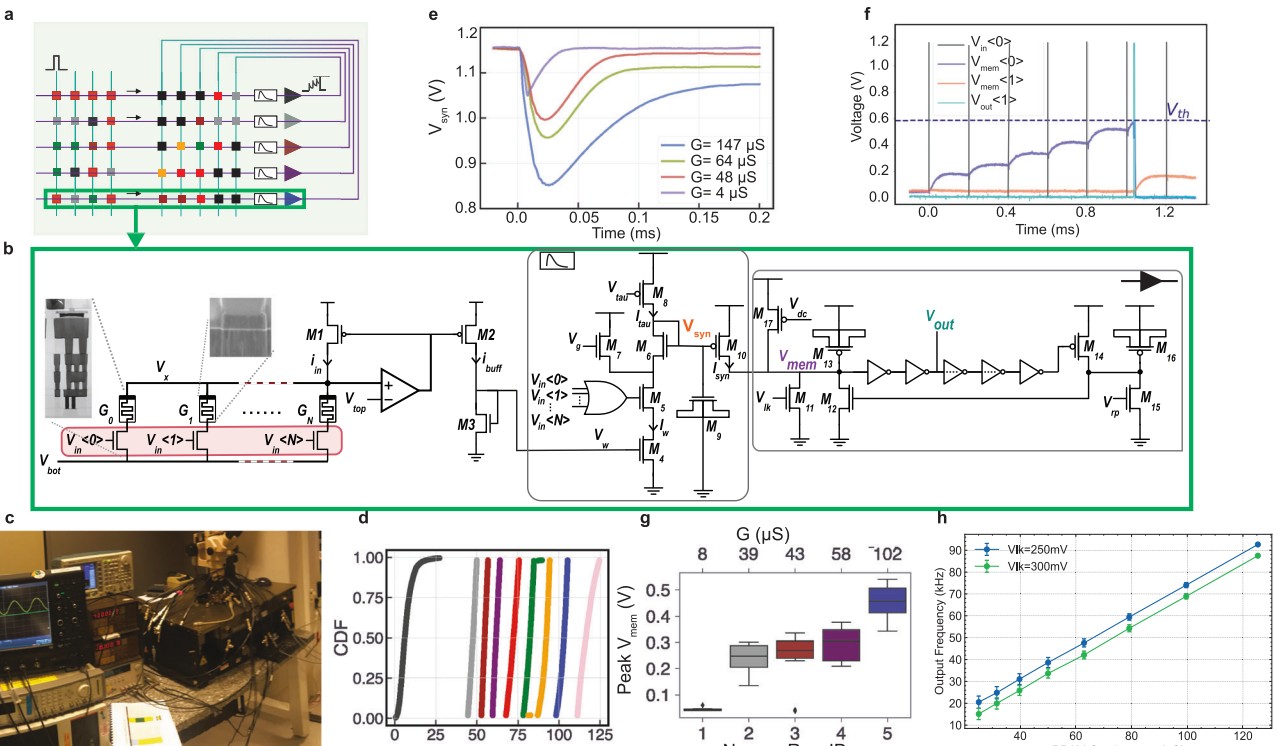

**Fig. 2 | Experimental results from the neuron column circuit. a** Neuron Tile, a crossbar with feed-forward and recurrent inputs displaying network parameters represented by colored squares. **b** Schematic of a single row of the fabricated crossbars, where RRAMs represent neuron weights. Insets show scanning and transmission electron microscopy images of the 1T1R stack with a hafnium-dioxide layer sandwiched between memristor electrodes. Upon input events $V_{in<i>}$, $V_{top} - V_{bot}$ is applied across $G_i$, yielding $i_{in}$ and subsequently $i_{buff}$, which feeds into the synaptic dynamic block, producing exponentially decaying current $i_{syn}$, with a time constant set by MOS capacitor $M_9$ and bias current $I_{tau}$. Integration of $i_{syn}$ into neuron membrane potential $V_{mem}$ triggers an output pulse ($V_{out}$) upon exceeding the inverter threshold. Refractory period regulation is achieved through MOS cap $M_{16}$ and $V_{rp}$ bias. **c** Wafer-level measurement setup utilizes an Arduino for logic

circuitry management to program RRAMs and a B1500 Device Parameter Analyzer to read device conductance. **d** Cumulative distributions of RRAM conductance ($G$) resulting from iterative programming in a 4096-device RRAM array with varied SET programming currents. **e** $V_{syn}$ initially at 1.2,V decreases as capacitor $M_9$ discharges upon pulse arrival at time 0. Discharge magnitude depends on $I_w$ set by $G$. Four conductance values' $V_{syn}$ curves are recorded. **f** Input pulse train (gray pulses) at $V_{in<0>}$ increases zeroth neuron's $V_{mem}$ (purple trace) until it fires (light blue trace) after six pulses, causing feedback influence on neuron 1's $V_{mem}$. **g** Statistical measurements on peak membrane potential in response to a pulse across a 5-neuron array over 10 cycles. **h** Neuron output frequency linearly correlates with $G$, with error bars reflecting variability across 4096 devices.

value is decided based on the "Threshold" conductance boundary in Fig. 3d. The Routing Tile regenerates a spike if the resulting $i_{buff}$ is higher than $i_{ref}$, and blocks it otherwise, since the output remains at zero. Therefore, the state of the device serves to either pass or block input spikes arriving from different input ports ($N, S, W, E$), sending them to its output ports (Supplementary Note 4). Since the routing array acts as a binary pass or no-pass, the decision boundary is on whether the devices is in its HCS or Low-Conductive State (LCS), as shown in Fig. 3d[46]. Using a fabricated Routing Tile circuit, we demonstrate its functionality experimentally in Fig. 3e. Continuous and dashed blue traces show the waveforms applied to the <N> and <S> inputs of the tile respectively, while the orange trace shows the response of the output towards the E port. The E output port follows the N input resulting from the corresponding RRAM programmed into its HCS, while the input from the S port gets blocked as the corresponding RRAM device is in its LCS, and thus the output remains at zero. This output pulse propagates onward to the next tile. Note that in Fig. 3d the output spike does not appear as rectangular due to the large capacitive load of the probe station (see Methods). To allow for greater reconfigurability, more channels per direction can be used in the Routing Tiles (see Supplementary Note 6).

## Hardware-aware simulations
**Application to real-time sensory-motor processing through hardware-aware simulations.** The Mosaic is a programmable

hardware well suited for the application of pre-trained small-world Recurrent Spiking Neural Network (RSNN) in energy and memory-constrained applications at the edge. Through hardware-aware simulations, we assess the suitability of the Mosaic on a series of representative sensory processing tasks, including anomaly detection in heartbeat (application in wearable devices), keyword spotting (application in voice command), and motor system control (application in robotics) tasks (Fig. 4a, b, c respectively). We apply these tasks to three network cases, (*i*) a non-constrained RSNN with full-bit precision weights (32 bit Floating Point (FP32)) (Fig. 4d), (*ii*) Mosaic constrained connectivity with FP32 weights (Fig. 4e), and (*iii*) Mosaic constrained connectivity with noisy and quantized RRAM weights (Fig. 4f). Therefore, case (*iii*) is fully hardware-aware, including the architecture choices (e.g., number of neurons per Neuron Tile), connectivity constraints, noise and quantization of weights.

For training case *i*, we use Backpropagation Through Time (BPTT)[47] with surrogate gradient approximations of the derivative of a LIF neuron activation function on a vanilla RSNN[48] (see Methods). For training case (*ii*), we introduce a Mosaic-regularized cost function during the training, which leads to a learned weight matrix with small-world connectivity that is mappable onto the Mosaic (see Methods). For case (*iii*), we quantize the weights using a mixed hardware-software experimental methodology whereby memory elements in a Mosaic software model are assigned conductance values programmed into a corresponding memristor in a fabricated array. Programmed

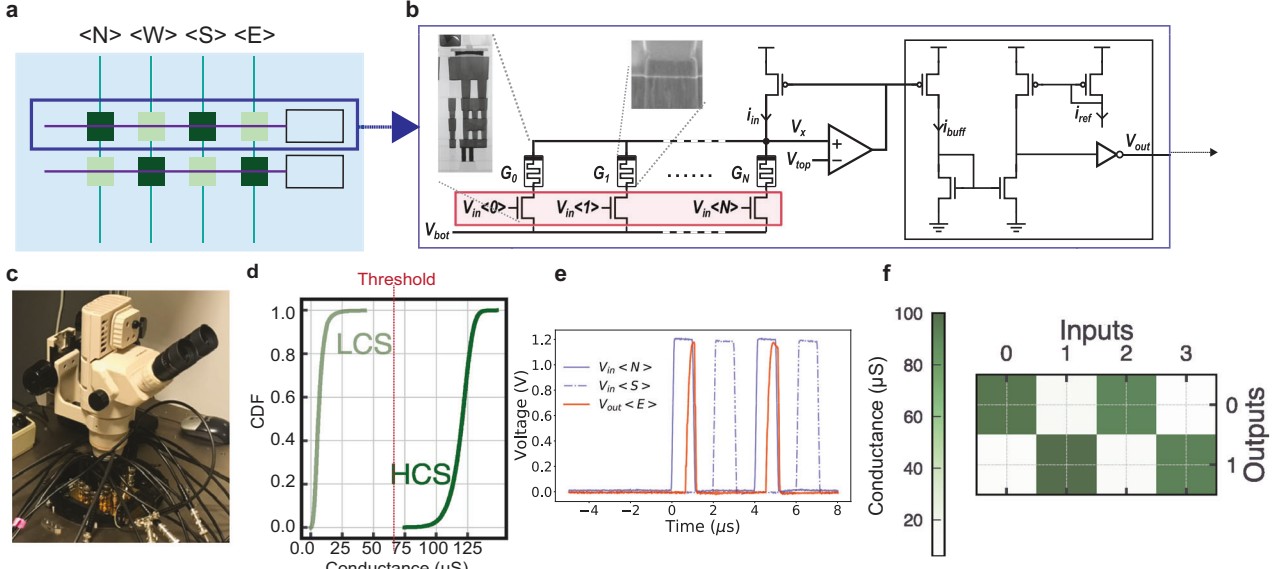

**Fig. 3 | Experimental measurements of the fabricated Routing Tile circuits.**
**a** The Routing Tile, a crossbar whose memory state steers the input spikes from different directions towards the destination. **b** Detailed schematic of one row of the fabricated routing circuits. On the arrival of a spike to any of the input ports of the Routing Tile, $V_{in} < i >$, a current proportional to $G_i$ flows in $i_{in}$, similar to the Neuron Tile. A current comparator compares this current against a reference current, $I_{ref}$, which is a bias current generated on chip through providing a DC voltage from the I/O pads to the gate of an on-chip transistor. If $i_{in} > i_{ref}$, the spike will get regenerated, thus "pass", or is "blocked" otherwise. **c** Wafer-level measurements of the test circuits through the probe station test setup. **d** Measurements from 4 kb array shows the Cumulative Distribution Function (CDF) of the RRAM in its High Conductive State (HCS) and Low Conductive State (LCS). The line between the distributions that separates the two is considered as the "Threshold" conductance, which the decision boundary for passing or blocking the spikes. Based on this Threshold value, the $I_{ref}$ bias in panel **b** is determined. **e** Experimental results from the Routing Tile, with continuous and dashed blue traces showing the waveforms applied to the < N > and < S > inputs, while the orange trace shows the response of the output towards the < E > port. The < E > output port follows the < N > input, resulting from the device programmed into the HCS, while the input from the < S > port gets blocked as the corresponding RRAM device is in its LCS. **f** A binary checker-board pattern is programmed into the routing array, to show a ratio of 10 between the High Resistive and Low Resistive state, which sets a perfect boundary for a binary decision required for the Routing Tile.

conductances are obtained through a closed-loop programming strategy[44,49–51].

For all the networks and tasks, the input is fed as a spike train and the output class is identified as the neuron with the highest firing rate. The RSNN of case (*i*) includes a standard input layer, recurrent layer, and output layer. In the Mosaic cases (*ii*) and (*iii*), the inputs are directly fed into the Mosaic Neuron Tiles from the top left, are processed in the small-world RSNN, and the resulting output is taken directly from the opposing side of the Mosaic, by assigning some of the Neuron Tiles in the Mosaic as output neurons. As the inputs and outputs are part of the Mosaic fabric, this scheme avoids the need for explicit input and output readout layers, relative to the RSNN, that may greatly simplify a practical implementation.

**Electrocardiography (ECG) anomaly detection.** We first benchmark our approach on a binary classification task in detecting anomalies in the ECG recordings of the MIT-BIH Arrhythmia Database[52]. In order for the data to be compatible with the RSNN, we first encode the continuous ECG time-series into trains of spikes using a delta-modulation technique, which describes the relative changes in signal magnitude[53,54] (see Methods). An example heartbeat and its spike encoding are plotted in Fig. 4a.

The accuracy over the test set for five iterations of training, transfer, and test for cases (*i*) (red), (*ii*) (green) and (*iii*) (blue) is plotted in Fig. 4g using a boxplot. Although the Mosaic constrains the connectivity to follow a small-world pattern, the median accuracy of case (*ii*) only drops by 3% compared to the non-constrained RSNN of case (*i*). Introducing the quantization and noise of the RRAM devices in case (*iii*), drops the median accuracy further by another 2%, resulting in a median accuracy of 92.4%. As often reported, the variation in the accuracy of case *iii* also increases due to the cycle-to-cycle variability of RRAM devices[51].

**Keyword spotting (KWS).** We then benchmarked our approach on a 20-class speech task using the Spiking Heidelberg Digit (SHD)[55] dataset. SHD includes the spoken digits between zero and nine in English and German uttered by 12 speakers. In this dataset, the speech signals have been encoded into spikes using a biologically-inspired cochlea model which effectively computes a spectrogram with Mel-spaced filter banks, and convert them into instantaneous firing rates[55].

The accuracy over the test set for five iterations of training, transfer, and test for cases (*i*) (red), (*ii*) (green) and (*iii*) (blue) is plotted in Fig. 4h using a boxplot. The dashed red box is taken directly from the SHD paper[55]. The Mosaic connectivity constraints have only an effect of about 2.5% drop in accuracy, and a further drop of 1% when introducing RRAM quantization and noise constraints. Furthermore, we experimented with various numbers of Neuron Tiles and the number of neurons within each Neuron Tile (Supplementary Note 8, Supplementary Fig. S10), as well as sparsity constraints (Supplementary Note 8, Supplementary Fig. S11) as hyperparameters. We found that optimal performance can be achieved when an adequate amount of neural resources are allocated for the task.

**Motor control by reinforcement learning.** Finally, we also benchmark the Mosaic in a motor system control Reinforcement Learning (RL) task, i.e., the Half-cheetah[56]. RL has applications ranging from active sensing via camera control[57] to dexterous robot locomotion[58].

To train the network weights, we employ the evolutionary strategies (ES) from Salimans et al.[59] in reinforcement learning settings[60–62]. ES enables stochastic perturbation of the network parameters,

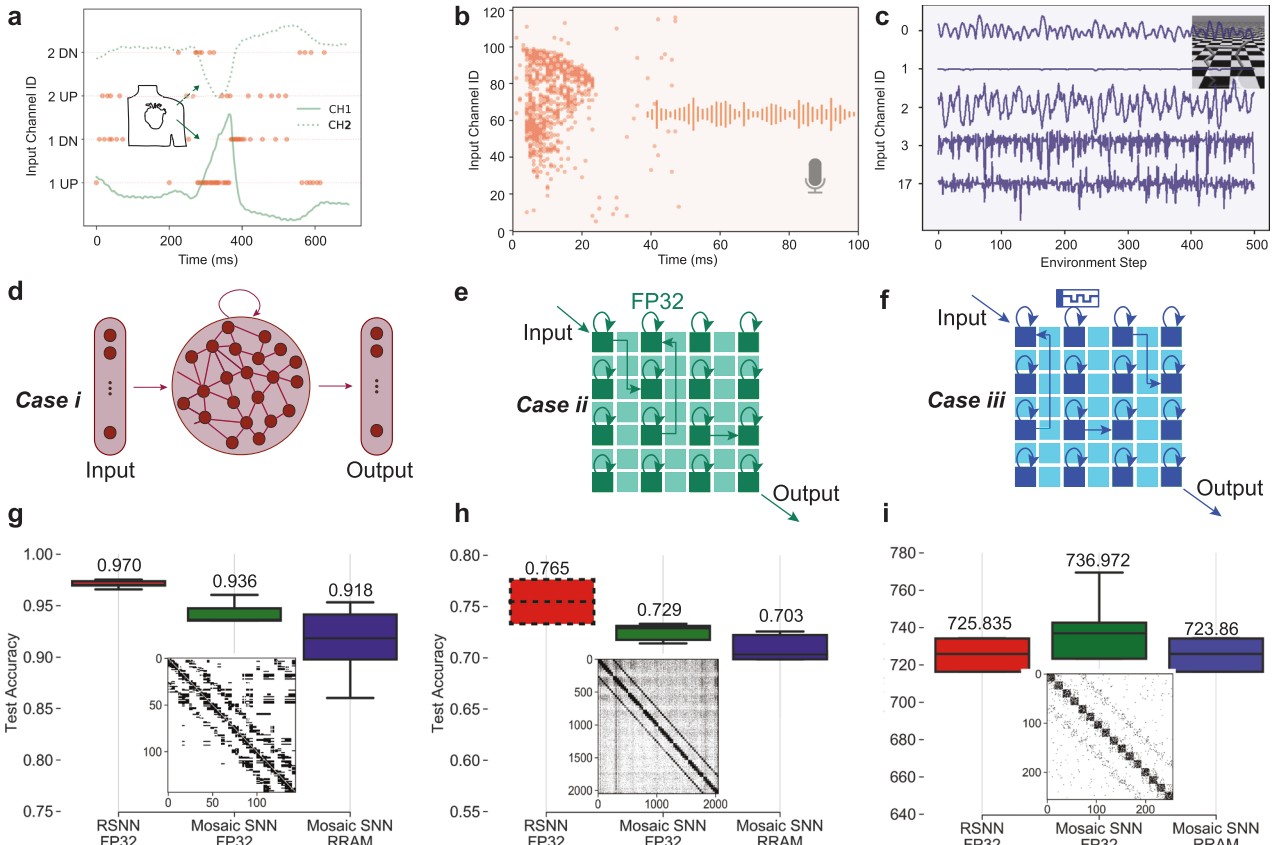

**Fig. 4 | Benchmarking the Mosaic against three edge tasks, heartbeat (ECG) arrhythmia detection, keyword spotting (KWS), and motor control by reinforcement learning (RL). a–c** A depiction of the three tasks, along with the corresponding input presented to the Mosaic. **a** ECG task, where each of the two-channel waveforms is encoded into up (UP) and down (DN) spiking channels, representing the signal derivative direction. **b** KWS task with the spikes representing the density of information in different input (frequency) channels. **c** half-cheetah RL task with input channels representing state space, consisting of positional values of different body parts of the cheetah, followed by the velocities of those individual parts. **d–f** Depiction of the three network cases applied to each task. **d** Case (*i*) involves a non-constrained Recurrent Spiking Neural Network (RSNN) with full-bit precision weights (FP32), encompassing an input layer, a recurrent layer, and an output layer. **e** Case (*ii*) represents Mosaic-constrained connectivity with FP32 weights, omitting explicit input and output layers. Input

directly enters the Mosaic, and output is extracted directly from it. Circular arrows denote local recurrent connections, while straight arrows signify sparse global connections between cores. **f** Case (*iii*) is similar to case (*ii*), but with noisy and quantized RRAM weights. **g–i** A comparison of task accuracy among the three cases: case (*i*) (red, leftmost box), case (*ii*) (green, middle box), and case (*iii*) (blue, right box). Boxplots display accuracy/maximum reward across five iterations, with boxes spanning upper and lower quartiles while whiskers extend to maximum and minimum values. Median accuracy is represented by a solid horizontal line, and the corresponding values are indicated on top of each box. The dashed red box for the KWS task with FP32 RSNN network is included from Cramer et al.[55] with 1024 neurons for comparison (with mean value indicated). This comparison reveals that the decline in accuracy due to Mosaic connectivity and further due to RRAM weights is negligible across all tasks. The inset figures depict the resulting Mosaic connectivity after training, which follows a small-world graphical structure.

evaluation of the population fitness on the task and updating the parameters using stochastic gradient estimate, in a scalable way for RL.

Figure 4i shows the maximum gained reward for five runs in cases i, ii, and iii, which indicates that the network learns an effective policy for forward running. Unlike tasks a and b, the network connectivity constraints and parameter quantization have relatively little impact.

Encouragingly, across three highly distinct tasks, performance was only slightly impacted when passing from an unconstrained neural network topology to a noisy small-world neural network. In particular, for the half-cheetah RL task, this had no impact.

## Neuromorphic platform routing energy

In-memory computing greatly reduces the energy consumption inherent to data movement in Von Neumann architectures. Although crossbars bring memory and computing together, when neural networks are scaled up, neuromorphic hardware will require an array of distributed crossbars (or cores) when faced with physical constraints, such as IR drop and capacitive charging[28]. Small-world networks may naturally permit the minimization of communication between these crossbars, but a certain energy and latency cost associated with the

data movement will remain, since the compilation of the small-world network on a general-purpose routing architecture is not ideal. Hardware that is specifically designed for small-world networks will ideally minimize these energy and latency costs (Fig. 1g). In order to understand how the spike routing efficiency of the Mosaic compares to other SNN hardware platforms, optimized for other metrics such as maximizing connectivity, we compare the energy and latency of (i) routing one spike within a core (0-hop), (ii) routing one spike to the neighboring core (1-hop) and (iii) the total routing power consumption required for tasks A and B, i.e., heartbeat anomaly detection and spoken digit classification respectively (Fig. 4a, b).

The results are presented in Table 1. We report the energy and latency figures, both in the original technology where the systems are designed, and scaled values to the 130 nm technology, where Mosaic circuits are designed, using general scaling laws[63]. The routing power estimates for Tasks A and B are obtained by evaluating the 0- and 1-hop routing energy and the number of spikes required to solve the tasks, neglecting any other circuit overheads. In particular, the optimization of the sparsity of connections between neurons implemented to train Mosaic assures that 95% of the spikes are routed with 0-hops

**Table 1 | Comparison of spike-routing performance across neuromorphic platforms**

| Neuromorphic chip | | TrueNorth[78] | SpiNNaker[79] | Neurogrid[80] | Dynap-SE[37] | Loihi[81] | Mosaic |
|---|---|---|---|---|---|---|---|
| Technology | | 28 nm (0.775 V) | 130 nm (1.2 V) | 180 nm (3 V) | 180 nm (1.8 V) | 14 nm (0.75 V) | 130 nm (1.2 V) |
| Routing | | On-chip | On-chip | On/off-chip | On-chip | On-chip | On-chip |
| 0-hop[b] energy | Original | 26 pJ | 30.3 nJ | 1 nJ | 30 pJ | 23.6 pJ | 400 fJ[a] |
| | sct.[c] 130 nm | 62.4 pJ | 30.3 nJ | 160 pJ | 13.4 pJ | 60.416 pJ | 400 fJ |
| 1-hop[d] energy | Original | 2.3 pJ | 1.11 nJ | 14 nJ | 17 pJ (@1.3V) | 3.5 pJ | 1.6 pJ[a] |
| | sct. 130 nm | 5.52 pJ | 1.11 nJ | 8.35 nJ | 17 pJ | 10.24 pJ | 1.6 pJ[a] |
| 1-hop latency | Original | 6.25 ns | 200 ps | 20 ns | 40 ns | 6.5 ns | 25 ns |
| | sct. 130 nm | 29 ns | 200 ps | 14.4 ns | 28.88 ns | 60.35 ns | 25 ns |
| Optimized for Small-Worldness | | No | No | No | Yes | No | Yes |
| Routing Power for task A | | 8.47 nW | 9.85 μW | 563.31 nW | 10.02 nW | 7.71 nW | 809 pW[a] |
| Routing Power for task B | | 272.82 nW | 317.08 μW | 18.14 μW | 322.7 nW | 248.41 nW | 5.06 nW[a] |

[a]Assuming an average resistance value of 10 kΩ, and a read pulse 10 ns width.
[b]The same as energy per Synaptic Operation (SOP), numbers are taken from Basu et al.[82].
[c]*sct.* Scaled to.
[d]Numbers are taken from Moradi et al.[37].

operations, while about 4% of the spikes are routed via 1-hop operations. The remaining spikes require k-hops to reach the destination Neuron Tile. The Routing energy consumption for Tasks A and B is estimated accounting for the total spike count and the routing hop partition.

The scaled energy figures show that although the Mosaic's design has not been optimized for energy efficiency, the 0- and 1-hop routing energy is reduced relative to other approaches, even if we compare with digital approaches in more advanced technology nodes. This efficiency can be attributed to the Mosaic's in-memory routing approach resulting in low-energy routing memory access which is distributed in space. This reduces (i) the size of each router, and thus energy, compared to larger centralized routers employed in some platforms, and (ii) it avoids the use of Content Addressable Memorys (CAMs), which consumes the majority of routing energy in some other spike-based routing mechanisms (Supplementary Note 2).

Neuromorphic platforms have each been designed to optimize different objectives[64], and the reason behind the energy efficiency of Mosaic in communication lies behind its explicit optimization for this very metric, thanks to its small-world connectivity layout. Despite this, as was shown in Fig. 4, the Mosaic does not suffer from a considerable drop in accuracy, at least for problem sizes of the sensory processing applications on the edge. This implies that for these problems, large connectivity between the cores is not required, which can be exploited for reducing the energy.

The Mosaic's latency figure per router is comparable to the average latency of other platforms. Often, and in particular, for neural networks with sparse firing activity, this is a negligible factor. In applications requiring sensory processing of real-world slow-changing signals, the time constants dictating how quickly the model state evolves will be determined by the size of the $V_{lk}$ in Fig. 2, typically on the order of tens or hundreds of milliseconds. Although, the routing latency grows linearly with the number of hops in the networks, as shown in the final connectivity matrices of Fig. 4g–i, the number of non-local connections decays down exponentially. Therefore, the latency of routing is always much less than the the time scale of the real-world signals on the edge, which is our target application.

The two final rows of Table 1 indicates the power consumption of the neuromorphic platforms in tasks A and B respectively. All the platforms are assumed to use a core (i.e., neuron tile) size of 32 neurons, and to have an N-hop energy cost equal to N times the 1-hop value. The potential of the Mosaic is clearly demonstrated, whereby a power consumption of only a few hundreds of pico Watts is required,

relative to a few nano/microwatts in the other neuromorphic platforms.

## Discussions

We have identified small-world graphs as a favorable topology for efficient routing, have proposed a hardware architecture that efficiently implements it, designed and fabricated memristor-based building blocks for the architecture in 130 nm technology, and report measurements and comparison to other approaches. We empirically quantified the impact of both the small-world neural network topology and low memristor precision on three diverse and challenging tasks representative of edge-AI settings. We also introduced an adapted machine learning strategy that enforces small-worldness and accounted for the low-precision of noisy RRAM devices. The results achieved across these tasks were comparable to those achieved by floating point precision models with unconstrained network connectivity.

Although the connectivity of the Mosaic is sparse, it still requires more number of routing nodes than computing nodes. However, the Routing Tiles are more compact than the neuron tiles, as they only perform binary classification. This means that the read-out circuitry does not require a large Signal to Noise Ratio (SNR), compared to the neuron tiles. This loosened requirement reduces the overhead of the Routing Tiles readout in terms of both area and power (Supplementary Note 9).

In this work, we have treated Mosaic as a standard RSNN, and trained it with BPTT using the surrogate gradient approximation, and simply added the loss terms that punish the network's dense connectivity to shape sparse graphs. Therefore, the potential computational advantages of small-world architectures do not necessarily emerge, and the performance of the network is mainly related to its number of parameters. In fact, we found that Mosaic requires more neurons, but about the same number of parameters, for the same accuracy as that of an RSNN on the same task. This confirms that taking advantage of small-world connectivity requires a novel training procedure, which we hope to develop in the future. Moreover, in this paper, we have benchmarked the Mosaic on sensory processing tasks and have proposed to take advantage of the small-worldness for energy savings thanks to the locality of information processing. However, from a computational perspective, these tasks do not necessarily take advantage of the small-wordness. In future works, one can foresee tasks that can exploit the small-world connectivity from a computational standpoint.

Mosaic favors local processing of input data, in contrast with conventional deep learning algorithms such as Convolutional and Recurrent Neural Networks. However, novel approaches in deep learning, e.g., Vision Transformers with Local Attention[65] and MLP-mixers[66], treat input data in a similar way as the Mosaic, subdividing the input dimensions, and processing the resulting patches locally. This is also similar to how most biological system processes information in a local fashion, such as the visual system of fruit flies[67].

In the broader context, graph-based computing is currently receiving attention as a promising means of leveraging the capabilities of SNNs[68–70]. The Mosaic is thus a timely dedicated hardware architecture optimized for a specific type of graph that is abundant in nature and in the real-world and that promises to find application at the extreme-edge.

## Methods

### Design, fabrication of Mosaic circuits

**Neuron and routing column circuits.** Both neuron and routing column share the common front-end circuit which reads the conductances of the RRAM devices. The RRAM bottom electrode has a constant DC voltage $V_{bot}$ applied to it and the common top electrode is pinned to the voltage $V_x$ by a rail-to-rail operational amplifier (OPAMP) circuit. The OPAMP output is connected in negative feedback to its non-inverting input (due to the 180 degrees phase-shift between the gate and drain of transistor $M_1$ in Fig. 2) and has the constant DC bias voltage $V_{top}$ applied to its inverting input. As a result, the output of the OPAMP will modulate the gate voltage of transistor $M_1$ such that the current it sources onto the node $V_x$ will maintain its voltage as close as possible to the DC bias $V_{top}$. Whenever an input pulse $V_{in}<n>$ arrives, a current $i_{in}$ equal to $(V_x - V_{bot})G_n$ will flow out of the bottom electrode. The negative feedback of the OPAMP will then act to ensure that $V_x = V_{top}$, by sourcing an equal current from transistor $M_1$. By connecting the OPAMP output to the gate of transistor $M_2$, a current equal to $i_{in}$, will therefore also be buffered, as $i_{buff}$, into the branch composed of transistors $M_2$ and $M_3$ in series. In the Routing Tile, this current is compared against a reference current, and if higher, a pulse is generated and transferred onwards. The current comparator circuit is composed of two current mirrors and an inverter (Fig. 3b). In the neuron column, this current is injected into a CMOS differential-pair integrator synapse circuit model[71] which generates an exponentially decaying waveform from the onset of the pulse with an amplitude proportional to the injected current. Finally, this exponential current is injected onto the membrane capacitor of a CMOS leaky-integrate and fire neuron circuit model[72] where it integrates as a voltage (see Fig. 2b). Upon exceeding a voltage threshold (the switching voltage of an inverter) a pulse is emitted at the output of the circuit. This pulse in turn feeds back and shunts the capacitor to ground such that it is discharged. Further circuits were required in order to program the device conductance states. Notably, multiplexers were integrated on each end of the column in order to be able to apply voltages to the top and bottom electrodes of the RRAM devices.

A critical parameter in both Neuron and Routing Tiles is the spike's pulse width. Minimizing the width of spikes assures maximal energy efficiency, but that comes at a cost. If the duration of the voltage pulse is too low, the readout current from the 1T1R will be imprecise, and parasitic effects due to the metal lines in the array might even impede the correct propagation of either the voltage pulse or the readout current. For this reason, we thoroughly investigated the minimal pulse-width that allows spikes and readout currents to be reliably propagated, at a probability of 99.7% ($3\sigma$). Extensive Monte Carlo simulation resulted in a spike pulse width of around 100 ns. Based on these SPICE simulations, we also estimated the energy consumption of Mosaic for the different tasks presented in Fig. 4.

**Fabrication/integration.** The circuits described in the Results section have been taped-out in 130 nm technology at CEA-Leti, in a 200 mm production line. The Front End of the Line, below metal layer 4, has been realized by ST-Microelectronics, while from the fifth metal layer upwards, including the deposition of the composites for RRAM devices, the process has been completed by CEA-Leti. RRAM devices are composed of a 5 nm thick $HfO_2$ layer sandwiched by two 5 nm thick $TiN$ electrodes, forming a $TiN/HfO_2/Ti/TiN$ stack. Each device is accessed by a transistor giving rise to the 1T1R unit cell. The size of the access transistor is 650 nm wide. 1T1R cells are integrated with CMOS-based circuits by stacking the RRAM cells on the higher metal layers. In the cases of the neuron and Routing Tiles, 1T1R cells are organized in a small - either $2 \times 2$ or $2 \times 4$ - matrix in which the bottom electrodes are shared between devices in the same column and the gates shared with devices in the same row. Multiplexers operated by simple logic circuits enable to select either a single device or a row of devices for programming or reading operations. The circuits integrated into the wafer, were accessed by a probe card which connected to the pads of the dimension of $[50 \times 90]\mu m^2$.

## RRAM characteristics

Resistive switching in the devices used in our paper are based on the formation and rupture of a filament as a result of the presence of an electric field that is applied across the device. The change in the geometry of the filament results in different resistive state in the device. A SET/RESET operation is performed by applying a positive/negative pulse across the device which forms/disrupts a conductive filament in the memory cell, thus decreasing/increasing its resistance. When the filament is formed, the cell is in the HCS, otherwise the cell is is the LCS. For a SET operation, the bottom of the 1T1R structure is conventionally left at ground level, and a positive voltage is applied to the 1T1R top electrode. The reverse is applied in the RESET operation. Typical values for the SET operation are $V_{gate}$ in [0.9 – 1.3] V, while the $V_{top}$ peak voltage is normally at 2.0 V. For the RESET operation, the gate voltage is instead in the [2.75, 3.25] V range, while the bottom electrode is reaching a peak at 3.0 V. The reading operation is performed by limiting the $V_{top}$ voltage to 0.3 V, a value that avoids read disturbances, while opening the gate voltage at 4.5 V.

## Mosaic circuit measurement setups

The tests involved analyzing and recording the dynamical behavior of analog CMOS circuits as well as programming and reading RRAM devices. Both phases required dedicated instrumentation, all simultaneously connected to the probe card. For programming and reading the RRAM devices, Source Measure Units (SMU)s from a Keithley 4200 SCS machine were used. To maximize stability and precision of the programming operation, SET and RESET are performed in a quasi-static manner. This means that a slow rising and falling voltage input is applied to either the Top (SET) or Bottom (RESET) electrode, while the gate is kept at a fixed value. To the $V_{top}(t)$, $V_{bot}(t)$ voltages, we applied a triangular pulse with rising and falling times of 1 sec and picked a value for $V_{gate}$. For a SET operation, the bottom of the 1T1R structure is conventionally left at ground level, while in the RESET case the $V_{top}$ is equal to 0 V and a positive voltage is applied to $V_{bot}$. Typical values for the SET operation are $V_{gate}$ in [0.9–1.3] V, while the $V_{top}$ peak voltage is normally at 2.0 V. Such values allow to modulate the RRAM resistance in an interval of [5–30] $k\Omega$ corresponding to the HCS of the device. For the RESET operation, the gate voltage is instead in the [2.75, 3.25] V range, while the bottom electrode is reaching a peak at 3.0 V.

The LCS is less controllable than the HCS due to the inherent stochasticity related to the rupture of the conductive filament, thus the HRS level is spread out in a wider [80–1000] $k\Omega$ interval. The reading operation is performed by limiting the $V_{top}$ voltage to 0.3 V, a value that avoids read disturbances, while opening the gate voltage at 4.5 V.

Inputs and outputs are analog dynamical signals. In the case of the input, we have alternated two HP 8110 pulse generators with a Tektronix AFG3011 waveform generator. As a general rule, input pulses had a pulse width of 1 μs and rise/fall time of 50 ns. This type of pulse is assumed as the stereotypical spiking event of a Spiking Neural Network. Concerning the outputs, a 1 GHz Teledyne LeCroy oscilloscope was utilized to record the output signals.

## Mosaic layout-aware training via regularizing the loss function

We introduce a new regularization function, $L_M$, that emphasizes the realization cost of short and long-range connections in the Mosaic layout. Assuming the Neuron Tiles are placed in a square layout, $L_M$ calculates a matrix $H \in \mathbb{R}^{j \times i}$, expressing the minimum number of Routing Tiles used to connect a source neuron $N_j$ to target neuron $N_i$, based on their Neuron Tile positions on Mosaic. Following this, a static mask $S \in \mathbb{R}^{j \times i}$ is created to exponentially penalize the long-range connections such that $S = e^{\beta H} - 1$, where $\beta$ is a positive number that controls the degree of penalization for connection distance. Finally, we calculate the $L_M = \sum S \odot W^2$, for the recurrent weight matrix $W \in \mathbb{R}^{j \times i}$. Note that the weights corresponding to intra-Neuron Tile connections (where $H = 0$) are not penalized, allowing the neurons within a Neuron Tile to be densely connected. During the training, task-related cross-entropy loss term (total reward in case of RL) increases the network performance, while $L_M$ term reduces the strength of the neural network weights creating long-range connections in Mosaic layout. Starting from the 10th epoch, we deterministically prune connections (replacing the value of corresponding weight matrix elements to 0) when their $L_1$-norm is smaller than a fixed threshold value of 0.005. This pruning procedure privileges local connections (i.e., those within a Neuron Tile or to a nearby Neuron Tile) and naturally gives rise to a small-world neural network topology. Our experiments found that gradient norm clipping during the training and reducing the learning rate by a factor of ten after 135th epoch in classification tasks help stabilize the optimization against the detrimental effects of pruning.

## RRAM-aware noise-resilient training

The strategy of choice for endowing Mosaic with the ability to solve real-world tasks is offline training. This procedure consists of producing an abstraction of the Mosaic architecture on a server computer, formalized as a Spiking Neural Network that is trained to solve a particular task. When the parameters of Mosaic are optimized, in a digital floating-point-32-bits (FP32) representation, they are to be transferred to the physical Mosaic chip. However, the parameters in Mosaic are constituted by RRAM devices, which are not as precise as the FP32 counterparts. Furthermore, RRAMs suffer from other types of non-idealities such as programming stochasticity, temporal conductance relaxation, and read noise[44,49–51].

To mitigate these detrimental effects at the weight transfer stage, we adapted the noise-resilient training method for RRAM devices[73,74]. Similar to quantization-aware training, at every forward pass, the original network weights are altered via additive noise (quantized) using a straight-through estimator. We used a Gaussian noise with zero mean and standard deviation equal to 5% of the maximum conductance to emulate transfer non-idealities. The profile of this additive noise is based on our RRAM characterization of an array of 4096 RRAM devices[44], which are programmed with a program-and-verify scheme (up to 10 iterations) to various conductance levels then measured after 60 seconds for modeling the resulting distribution.

## ECG task description

The Mosaic hardware-aware training procedure is tested on a electrocardiogram arrhythmia detection task. The ECG dataset was downloaded from the MIT-BIH arrhythmia repository[52]. The database is composed of continuous 30-min recordings measured from multiple subjects. The QRS complex of each heartbeat has been annotated as either healthy or exhibiting one of many possible heart arrhythmias by a team of cardiologists. We selected one patient exhibiting approximately half healthy and half arrhythmic heartbeats. Each heartbeat was isolated from the others in a 700 ms time-series centered on the labeled QRS complex. Each of the two 700 ms channel signals were then converted to spikes using a delta modulation scheme[75]. This consists of recording the initial value of the time-series and, going forward in time, recording the time-stamp when this signal changes by a pre-determined positive or negative amount. The value of the signal at this time-stamp is then recorded and used in the next comparison forward in time. This process is then repeated. For each of the two channels this results in four respective event streams - denoting upwards and downwards changes in the signals. During the simulation of the neural network, these four event streams corresponded to the four input neurons to the spiking recurrent neural network implemented by the Mosaic.

Data points were presented to the model in mini-batches of 16. Two populations of neurons in two Neuron Tiles were used to denote whether the presented ECG signals corresponded to a healthy or an arrhythmic heartbeat. The softmax of the total number of spikes generated by the neurons in each population was used to obtain a classification probability. The negative log-likelihood was then minimized using the categorical cross-entropy with the labels of the signals.

## Keyword spotting task description

For keyword spotting task, we used SHD dataset (20 classes, 8156 training, 2264 test samples). Each input example drawn from the dataset is sampled three times along the channel dimension without overlap to obtain three augmentations of the same data with 256 channels each. The advantage of this method is that it allows feeding the input stream to fewer Neuron Tiles by reducing the input dimension and also triples the sizes of both training and testing datasets.

We set the simulation time step to 1 ms in our simulations. The recurrent neural network architecture consists of 2048 LIF neurons with 45 ms membrane time constant. The neurons are distributed into $8 \times 8$ Neuron Tiles with 32 neurons each. The input spikes are fed only into the neurons of the Mosaic layout's first row (8 tiles). The network prediction is determined after presenting each speech data for 100 ms by counting the total number of spikes from 20 neurons (total number of classes) in 2 output Neuron Tiles located in the bottom-right of the Mosaic layout. The neurons inside input and output Neuron Tiles are not recurrently connected. The network is trained using BPTT on the loss $L = L_{CE} + \lambda L_M$, where $L_{CE}$ is the cross-entropy loss between input logits and target and $L_M$ is the Mosaic-layout aware regularization term. We use batch size of 512 and suitably tuned hyperparameters.

## Reinforcement learning task description

In the RL experiments, we test the versatility of a Mosaic-optimized RSNN on a continuous action space motor-control task, half-cheetah, implemented using the BRAX physics engine for rigid body simulations[76]. At every timestep $t$, environment provides an input observation vector $o^t \in \mathbb{R}^{25}$ and a scalar reward value $r^t$. The goal of the agent is to maximize the expected sum of total rewards $R = \sum_{t=0}^{1000} r^t$ over an episode of 1000 environment interactions by selecting action $a^t \in \mathbb{R}^7$ calculated by the output of the policy network. The policy network of our agent consists of 256 recurrently connected LIF neurons, with a membrane decay time constant of 30 ms. The neuron placement is equally distributed into 16 Neuron Tiles to form a $7 \times 7$ Mosaic layout. We note that for simulation purposes, selecting a small network of 16 Neuron Tiles with 16 neurons each, while not optimal in terms of memory footprint (Eq. (2)), was preferred to fit the large ES population within the constraints of single GPU memory capacities. At each time step, the observation vector $o^t$ is accumulated into the membrane voltages of the first 25 neurons of two

upper left input tiles. Furthermore, action vector $a^t$ is calculated by reading the membrane voltages of the last seven neurons in the bottom right corner after passing *tanh* non-linearity.

We considered Evolutionary Strategies (ES) as an optimization method to adjust the RSNN weights such that after the training, the agent can successfully solve the environment with a policy network with only locally dense and globally sparse connectivity. We found ES particularly promising approach for hardware-aware training as (i) it is blind to non-differentiable hardware constraints e.g., spiking function, quantized weights, connectivity patterns, and (ii) highly parallelizable since ES does not require spiking variable to be stored for thousand time steps compared to BPTT that explicitly calculates the gradient. In ES, the fitness function of an offspring is defined as the combination of total reward over an episode, $R$ and realization cost of short and long-range connections $L_M$ (same as KWS task), such that $F = R - \lambda L_M$. We used the population size of 4096 (with antithetic sampling to reduce variance) and mutation noise standard deviation of 0.05. At the end of each generation, the network weights with $L_0$-norm smaller than a fixed threshold are deterministically pruned. The agent is trained for 1000 generations.

### Calculation of memory footprint

We calculate the Mosaic architecture's Memory Footprint (MF) in comparison to a large crossbar array, in building small-world graphical models.

To evaluate the MF for one large crossbar array, the total number of devices required to implement any possible connections between neurons can be counted - allowing for any Spiking Recurrent Neural Networks (SRNN) to be mapped onto the system. Setting $N$ to be the number of neurons in the system, the total possible number of connections in the graph is $MF_{ref} = N^2$.

For the Mosaic architecture, the number of RRAM cells (i.e., the MF) is equal to the number of devices in all the Neuron Tiles and Routing Tiles: $MF_{mosaic} = MF_{NeuronTiles} + MF_{RoutingTiles}$.

Considering each Neuron Tile with $k$ neurons, each Neuron Tile contributes to $5k^2$ devices (where the factor of 5 accounts for the four possible directions each tile can connect to, plus the recurrent connections within the tile). Evenly dividing the $N$ total number of neurons in each Neuron Tile gives rise to $T = ceil(N/k)$ required Neuron Tiles. This brings the total number of devices attributed to the Neuron Tile to $T \cdot 5k^2$.

The number of Routing Tiles that connects all the Neuron Tiles depends on the geometry of the Mosaic systolic array. Here, we assume Neuron Tiles assembled in a square, each with a Routing Tile on each side. We consider $R$ to be the number of Routing Tiles with $(4k)^2$ devices in each. This brings the total number of devices related to Routing Tiles up to $MF_{RoutingTiles} = R \cdot (4k)^2$.

The problem can then be re-written as a function of the geometry. Considering Fig. 1g, let $i$ be an integer and $(2i+1)^2$ the total number of tiles. The number of Neuron Tiles can be written as $T = (i+1)^2$, as we consider the case where Neuron Tiles form the outer ring of tiles. As a consequence, the number of Routing Tiles is $R = (2i+1)^2 - (i+1)^2$. Substituting such values in the previous evaluations of $MF_{NeuronTiles} + MF_{RoutingTiles}$ and remembering that $k < N \cdot T$, we can impose that $MF_{Mosaic} = MF_{NeuronTiles} + MF_{RoutingTiles} < MF_{MF_{ref}}$.

This results in the following expression:

$$MF_{Mosaic} = MF_{NeuronTiles} + MF_{RoutingTiles} < MF_{reference} \quad (1)$$

$$(i+1)^2 (5k^2) + [(2i+1)^2 - (i+1)^2](4k)^2 < (k(i+1)^2)^2 \quad (2)$$

This expression can then be evaluated for $i$, given a network size, giving rise to the relationships as plotted in Fig. 1g in the main text.

## Data availability

The MIT-BIH ECG dataset[52], the Spiking Heidelberg Datasets[77], and half-cheetah from OpenAI gym[56] are publicly accessible. All other measured data are freely available upon request.

## Code availability

The code is available on https://github.com/EIS-Hub/Mosaic.

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

## Acknowledgements
We acknowledge funding support from the H2020 MeM-Scales project (871371) (F.M., G.I., E.V., M.P.), Swiss National Science Foundation Starting Grant Project UNITE (TMSGI2-211461) (F.M., M.P.), Marie Skłodowska-Curie grant agreement No 861153 (Y.D., G.I.), as well as European Research Council consolidator grant DIVERSE (101043854) (E.V.). We are grateful to Emre Neftci, Shyam Narayanan, Junren Chen and Zhe Su for helpful discussions throughout the project.

## Author contributions
T.D., G.I., E.V. and M.P. developed the Mosaic concept. T.D. and M.P. designed and laid out the circuits for fabrication. The circuits were fabricated under the supervision of E.V. Characterization and verification of the fabricated circuits were done by F.M. and A.P. Hardware-aware training simulations and benchmarking was conducted by Y.D. and F.M. All authors contributed to writing of the manuscript. M.P. supervised the project.

## Competing interests
The authors declare no competing interests.
