## [Peer Review File · Nature Communications]

REVIEWER COMMENTS

Reviewer #1 (Remarks to the Author):

Thomas Dalgaty and colleagues have presented their work on employing brain-inspired sparsity in post-CMOS electronic circuits, to minimize wasting connections as is often done in more traditional crossbar architectures (which are widely used today). Further, many problems don't require dense connections. The idea of employing sparsity to improve connectivity, scalability and performance, and also the fact that many problems can be solved efficiently on sparse architectures, is not really new. However, I don't think this idea has been employed widely in CMOS-integrated memristor chips yet. To that end, the direction proposed by the authors is novel. On the downside, I am unclear on what has been experimentally demonstrated in this work, and how much of it is mostly a proposal. The team does have the full capability to develop small-scale versions of their proposed architecture, but it appears like they have only partial experimental results. Further, the quality of the performance benchmarks are unclear. My specific comments are below.

1. Figs. 1c-d are not clear. For instance, it needs a colorbar. Also, the inputs being represented as high/low memory makes it unclear. I believe that all the figure tries to show is that the near-diagonal weights are high, so only the near neighbors are strongly connected.
2. Fig. 1g - make the horizontal lines and the data matched in colors.
3. Unclear on Fig. 1f. Green units seem just as densely connected as panel d, which makes this approach appear to not improve in wasting many connections. What you actually want to show is likely that the green units can have arbitrary density. For that, see suggestion in comment (5).
4. Clarify Fig. S2. Why do you have outputs on the left and also the bottom? Similarly, in Fig. 8, why do you have the outputs feeding back into the inputs? Is this some kind of self feedback?
5. In Fig. 1, illustrate dense and sparse connections via two schematic examples. Each example would be a pair of a graph and three tiles (two weights and one connection). Similar to Fig. S2. Show both the dense and sparse illustrations for the same graph. Similar to Fig. S4, but for the same graph.
6. It's hard to tell what is experimental and what is simulated. In Fig. 4, and in Table 1, I suppose that everything is simulated? And you seem to have experimental results from a single tile (both a connecting tile and a weight tile, Figs. 2 and 3). Is this right? Within each tile, do you have statistical

distributions of the full tile, or is it just a single column? If it is the full tile, then please show the distributions, histograms, error statistics, etc., of programmed patterns. Looks like you don't have data for signal moving across the architecture (am I right?). If so, why not? Does your chip not consist of the full architecture, and only a single tile is in silicon? Were you able to study the signal moving experimentally between just three tiles (two weight and one neuron)?

7. I understand that Table 1 might be purely simulations or simple calculations. If so, just state it. However, I am not sure of what size is being compared in tasks A and B. Specifically, how are you accounting for all the communication, cooling, and other overheads that the other chips have to deal with? Are tasks A and B the size of an entire chip in the other cases? In other words, it would be unfair to compare a small problem being solved on your chip to the same small problem being solved on a GPU if the problem occupies only 1% of the GPU's circuits. At the least state this size, and be upfront about the limitations of your analysis (and its scale). It would also help to have an additional row in Table 1 showing the scale of each chip.

Finally, please include full details of how you arrived at your benchmark numbers for all the chips for each task, so that any reader can easily reproduce your analysis.

Reviewer #2 (Remarks to the Author):

This engaging paper thoroughly presents the implementation of a neuromorphic processor using RRAM (Resistive Random-Access Memory). Capitalizing on the small-world property, it effectively achieves tight coupling of neurons within a neural tile via internal connections, while maintaining loose coupling between neural tiles through RRAM-based routers. I have several questions about this paper and would appreciate the opportunity to discuss them with the authors.

1) The explanation of the routing tile mechanism in the paper was not entirely clear to me. What does the routed message entail? How is the routing algorithm determined and integrated into the router implemented in this paper?

2) Concerning the neuron tile, it appears to significantly differ from existing digital circuit-based systolic circuits. While input spikes can propagate to other neurons, this functionality seems achievable with conventional crossbar-based RRAM as well. Therefore, further clarification is needed to differentiate these two approaches.

3) How is the reference current, which determines whether an input spike passes through the router, established and provided? Who is responsible for this?

4) How are connections between the router tile and neuron tile formed? What characterizes the communication between them? (i.e., what constitutes the message?)

5) What technique was employed to measure the power consumption of the entire chip?

6) When making comparisons, how were the results for mosaic SNN FP32 obtained? Since RRAM cannot achieve high-precision FP32, if it was implemented using digital circuits, why is there a discrepancy between the results compared to software?

7) It would be helpful to provide specific data concerning the loss of precision for the three benchmarks.

8) Does this chip include any peripheral circuitry? How are input and output implemented?

In conclusion, I find this paper quite intriguing and look forward to further discussions with the authors.

This article by Dalgaty et al. reports on a new neuromorphic architecture, “Mosaic”, aimed at implementing small-world spiking neural networks efficiently. The architecture mainly consists of neuron tiles interspersed among routing tiles, allowing for dense intra-tile connectivity and sparse inter-tile connectivity. These tiles are designed using HfO₂ RRAM as either synaptic or routing matrices, alongside post-RRAM CMOS circuitries in the 130 nm technology. Three different tasks were simulated, achieving competitive performance figures.

The work presents a novel method of implementing SNNs of varying scales on hardware, which is an important topic. However, several comments and concerns need to be addressed.

1. It is not entirely clear what the purpose of Figure 2c is. This figure shows the CDF of conductance states programmed using a single SET pulse, but the RRAMs are programmed using a write-and-verify strategy when used in the task demonstrations. It does not reflect the actual weight transfer non-idealities, nor does it show the range and number of possible states. It might also create some confusion at first glance as it shows a ratio of <3 between the highest and the lowest states, which is very low. Also, it would be good if the authors provide more device- or array-level characterizations in addition to the CDF of conductance states.
2. It seems that the synaptic arrays only accept positive weights. The synaptic part of the neuron circuit does not support commonly used strategies such as differential conductance to realize negative weights. Can the authors clarify how should negative weights be mapped onto the proposed hardware?
3. Logically, when each neuron fires, the inter-tile spike outputs towards the 4 directions should be identical copies of the neuron output itself. Can the authors clarify this and make it clear in the main text? Also, Figure S2 gives the impression that the inter-tile outputs towards each of the 4 directions originate from different neurons, which creates confusion if the aforementioned case is true.
4. Is the size of each neuron tile and routing tile fixed during the fabrication of the entire hardware? If so, is there an optimal size for these tiles such that networks with a wide range of sizes can be efficiently implemented?
5. It is interesting to see that the Mosaic constrained network shows little to no performance degradation for the chosen constraining parameters. Nevertheless, did the authors experiment with further reducing the number of neurons per neuron tile to see the degree to which tighter constraints impact network performance? Could this serve as a guide to designing optimally sized tiles for the architecture?
6. The network performance should decrease with a smaller network scale. How does this trade-off of a Mosaic-constrained network compare to that of a conventional network? If only a certain number of memory units are allocated for a network, is it possible for a conventional RSNN (with fewer neurons) and a Mosaic-constrained network (with more neurons) to achieve similar performance figures? Ruling out such a possibility will further strengthen the argument that small-world networks and, accordingly, the proposed architecture is a better way of doing neuromorphic computing.
7. The workings of the Mosaic constrained network, in particular the input stage of the network, are not clear. Are all the elements of the input vector linearly transformed first

before being accumulated by the neurons, or is each input element only accumulated by their corresponding neuron? Figure S8 seems to support the former, but the description in the Methods section seems to support the latter. The concern here is that if the inputs span multiple tiles as in the case of the KWS and RL tasks, and there is an initial linear transformation, inputs into one tile cannot contribute (or can only weakly contribute) to the neuron state of other input-receiving tiles. The input channels are thus automatically grouped based on the tiles they are situated in and might contribute unequally to the network. In other words, if a set of input channels contains more useful information than others, then there might exist a way of grouping these input channels to maximize network performance. Also, for the KWS task, the neurons in the input tiles are not recurrently connected. Is it the same for the other tasks? Similarly, if these input neurons span multiple tiles and are still recurrently connected, they might also contribute unequally to the network.

8. It would be good if the authors provide more detail on how the energy/power figures of the architecture were obtained/calculated. Besides, the proposed architecture requires more routing tiles than neuron tiles, and a routing tile has 4x the number of post-array circuitries (op-amp, current comparator) than a neuron tile, all of which will add up to the total power consumption. How significant is this, especially when compared with using a simple crossbar array architecture? This is important as the deployment of the proposed hardware is aimed at extreme-edge applications, as mentioned by the authors.

Minor issues:

9. On page 8 line 202: “The final row of Table 1 indicates the power consumption of all of the hardware in task A (ECG anomaly detection).”
The second last row and final row refer to routing power for task A and task B, respectively.
10. On page 9 line 222: “(due to the 90 degrees phase-shift between the gate and drain of transistor M1 in Fig. 2)”
The phase shift between the gate and drain of M_1 should be 180° , not 90° .
11. On page 9 line 230:
Did the authors cite the wrong supplementary figure? If so, Figure S8 is not cited anywhere else in the main text.
12. On page 11:
From the schematic in Figures 1 and 2a, each neuron tile has dedicated intra-tile recurrent connections on top of inter-tile inputs from the 4 directions. So, is the total number of RRAM devices in each neuron tile $5k^2$ instead of $4k^2$ (k inputs from 4 directions contribute $4k^2$, recurrent connections contribute k^2)?
13. On page 11 line 365: “We consider R to be the number of routing tiles with $4k^2$ devices in each.”
 $4k^2$ should be $(4k)^2$?

Response to the reviewers:

Mosaic: in-memory computing and routing for small-world spike-based neuromorphic systems

We sincerely thank all the reviewers for their insightful comments on our manuscript. We believe addressing the comments has made the paper much stronger and clear, and we are grateful for that. Below, we reply to each concern of the reviewers one by one. We have addressed the changes in the main and supplementary texts, highlighted with a strike-through red when text was removed, and in blue when text was added. In this letter, we reply with black, whenever the changed text is copied here, we highlight it in green for your attention. Thank you very much.

Response to Reviewer 1

Thomas Dalgaty and colleagues have presented their work on employing brain-inspired sparsity in post-CMOS electronic circuits, to minimize wasting connections as is often done in more traditional crossbar architectures (which are widely used today). Further, many problems don't require dense connections. The idea of employing sparsity to improve connectivity, scalability and performance, and also the fact that many problems can be solved efficiently on sparse architectures, is not really new. However, I don't think this idea has been employed widely in CMOS-integrated memristor chips yet. To that end, the direction proposed by the authors is novel. On the downside, I am unclear on what has been experimentally demonstrated in this work, and how much of it is mostly a proposal. The team does have the full capability to develop small-scale versions of their proposed architecture, but it appears like they have only partial experimental results. Further, the quality of the performance benchmarks are unclear. My specific comments are below.

Reply 1 We thank the reviewer for acknowledging the novelty of our implementation, and for their time to go through our paper in details and bringing up very relevant points. We have clarified in the paper which parts are experimental or hardware aware simulations based on measurements. For the detailed response to your concerns, please see the point to point response below.

1. Figs. 1c-d are not clear. For instance, it needs a colorbar. Also, the inputs being represented as high/low memory makes it unclear. I believe that all the figure tries to show is that the near-diagonal weights are high, so only the near neighbors are strongly connected.

Reply 2

Thank you for pointing out the lack of clarity of the figure. Indeed, the point of the figure was to show that the near-diagonal connections are strong and the rest are weak.

[Action] We have changed Figure 1 based on your comments. You can see it in the main text, and we also copy it here for your convenience:

To make it more clear, we have made the following changes to Fig. 1c.

- We have replaced part c, which used to be a colored graph, with a black-and-white graph, explaining in the caption that the darkest color represents stronger connectivity.
- We mirrored the connectivity matrix, to match the mapping onto the crossbar array.
- We have drawn two corresponding red rectangles around the part with the strongest connectivity to highlight the mapping. Also, we have drawn two rectangles in blue, which shows which part of the memory crossbar array is not being used, and thus is "wasted".

2. Fig. 1g - make the horizontal lines and the data matched in colors.

Reply 3 Thank you for pointing this out. We have replotted it following your suggestion.

[Action] We have modified Figure 1g matching the color of the horizontal lines with that of the plotted lines.

Figure 1. Redrawn Fig.1 of the main text. For the changes in the caption, please refer to the diff file.

3. Unclear on Fig. 1f. Green units seem just as densely connected as panel d, which makes this approach appear to not improve in wasting many connections. What you actually want to show is likely that the green units can have arbitrary density. For that, see suggestion in comment (5).

Reply 4 Thank you very much for this. Please refer to our reply 2. Part f actually does want to point out that instead of a sparsely connected crossbar matrix in d, where most of the devices are wasted, with small-world connectivity, we will have a good utilization of a small crossbar array, which is densely connected.

[Action] To make the difference between panel d and f distinguishable, we have highlighted panel f with red rectangle and blue triangle, to show which part is wasted and which part is useful. We hope that this makes it more clear.

4. Clarify Fig. S2. Why do you have outputs on the left and also the bottom? Similarly, in Fig. 8, why do you have the outputs feeding back into the inputs? Is this some kind of self-feedback?

Reply 5 We are sorry that this was not clear. There is indeed self-recurrency in each neuron tile. The neuron tile receives feed-forward inputs from 4 directions (neighboring routers), and self-recurrent inputs from itself. The neurons integrate these inputs, and once the integration passes the threshold, they generate output spikes, which they send out to 4 directions. i.e., the same spikes gets copied in 4 directions, and depending on the state of the routers, they propagate in the 2D Mosaic.

[Action] We have completely re-drawn Fig. S2 (now S4), which now provides a lot more information about the details of each crossbar, and how information flows in/out of neuron and routing tiles. This is provided in the supplementary information, and a scaled version is also copied here for your convenience, along with the caption.

We have also provided more detailed explanation in the caption of S2, and Supplementary Note 3 on how the information flows in each crossbar and how it is spread in the network.

Figure 2. Neuron tiles (green) transfer information in the form of spikes to each other through routing tiles (blue). Details of the Mosaic architecture is shown with the size of the neuron and routing tiles. The neuron tiles receive feed-forward input from four directions of North (N), East (E), West (W), and South (S), and local recurrent input from the neurons in the tile. The neurons integrate the information and once spike, send their output to 4 directions. Having 4 neurons in a tile, gives rise to 16 outputs (4 outputs copied in 4 directions), and 20 inputs (4 inputs from 4 directions (16), plus 4 recurrent inputs). The routing tiles receive 16 inputs (4 inputs from 4 directions) and send out 16 outputs (4 outputs in 4 directions). In the crossbars, the red squares and black squares represent devices in their high conductive and low conductive state, respectively. The connection between the neuron tile and the routing tile is directly through a wire. For instance, $V_{out} < 3 : 0 >$ is the same as the $V_{in,W}$, and $V_{in,E} < 3 : 0 >$ is the same as $V_{out,W}$.

5. In Fig. 1, illustrate dense and sparse connections via two schematic examples. Each example would be a pair of a graph and three tiles (two weights and one connection). Similar to Fig. S2. Show both the dense and sparse illustrations for the same graph. Similar to Fig. S4, but for the same graph.

Reply 6 Dear Reviewer, we are happy to clarify the connectivity architecture of Mosaic. For this sake, we constructed two graphs, following the structure that you suggested, made of 2 Neuron Tiles and one Routing Tile. By controlling the probability of connections within the Neuron and Routing Tiles, we can produce a densely connected graph (left) with $p_{NT} = 0.75$, $p_{RT} = 0.6$, and a sparse graph (right) with $p_{NT} = 0.30$, $p_{RT} = 0.05$. We also added the schematics of the small Mosaic architecture where connections are drawn as black dots.

[Action] We have added a new Supplementary Note with the following figure, explaining how different graphical structures can be implemented using the three example tiles of the Mosaic architecture, as you suggested, and also showed its corresponding connectivity matrix. This is the new Supplementary Note 3, and new Fig. S3 which is copied here for your convenience:

Figure 3. Mosaic connectivity example, formed by setting the probability of connection within Neuron Tile (p_{NT}) and Routing Tiles (p_{RT}). (left) Densely connected Mosaic composed of 2 Neuron Tiles and 1 Routing Tile. The graph related to its connectivity is shown as well adjacency matrix. (right) Sparsely connected Mosaic. The graph is programmed to favor the intra-Neuron Tile connectivity and allow for two clusters to emerge, penalizing connections between the two clusters.

6. It's hard to tell what is experimental and what is simulated. In Fig. 4, and in Table 1, I suppose that everything is simulated? And you seem to have experimental results from a single tile (both a connecting tile and a weight tile, Figs. 2 and 3). Is this right? Within each tile, do you have statistical distributions of the full tile, or is it just a single column? If it is the full tile, then please show the distributions, histograms, error statistics, etc., of programmed patterns. Looks like you don't have data for signal moving across the architecture (am I right?). If so, why not? Does your chip not consist of the full architecture, and only a single tile is in silicon? Were you able to study the signal moving experimentally between just three tiles (two weights and one neuron)?

Reply 7 Indeed, Fig. 4 and Table 1 are simulations, and the experimental results are from individual test circuits implementing neuron and routing tiles that that we have designed and fabricated. The full architecture is our next step, and we are working towards building that during the upcoming year.

[Action] We have now made it more clear in the paper what is simulation and what is experimental, by introducing two subsections of “Hardware measurements” and “Hardware-aware simulations” to the paper. So it is now clear that the first part of the Results section is experimental, and the second part is simulations based on those measurements. We have added statistical distributions for neuron tiles and routing tiles in Fig. 2 and Fig. 3 respectively.

Fig. 2: In the neuron tile, we have made new statistical measurements on the membrane potential of the 5 neurons in the neuron tile, for different values of conductances. This is now shown in Fig. 2g. To make it clear that it is a measurement, we have added the picture of the measurement set up in the Fig. 2c. This is copied here for your convenience and is reflected in the main text. (Fig. 4 of this document).

Fig. 3: In the routing tile, we have made a binary checker board pattern, to show case a perfect boundary for a binary decision required for the routing tile, thanks to the ratio of 10 between the High Resistive and Low Resistive state (shown in the CDF plot). This is copied here for your convenience and is reflected in the main text. (Fig. 5 of this document).

Figure 4. Experimental results from the neuron column circuit. (a) The neuron tile, a crossbar with feed-forward and recurrent inputs, where the colorful squares hold the parameters of the network and their color corresponds to different weight values. (b) Detailed schematic of one row of the fabricated crossbars. RRAMs conductance represents the weights of the neurons. Insets of (left) scanning electron and (right) transmission electron microscopy images respectively show cross-sections of the 1T1R stack and the hafnium-dioxide layer sandwiched between top and bottom memristor electrodes. On the arrival of any of the input events $V_{in}$, read voltage equivalent to $V_{top} - V_{bot}$ is applied across G_i , giving rise to a current pulse i_{in} , and in turn to i_{buff} , feeding to the synaptic dynamic block. The “Differential Pair Integrator” low-pass filters the current pulse, giving rise to an exponentially decaying current dynamic, whose time constant is a function of the MOS capacitor M_9 and I_{tau} . The current is then fed and integrated by the neuron’s membrane potential, V_{mem} , and upon passing the threshold of the inverter creates an output pulse (V_{out}). The refractory period time is determined by the MOS cap M_{16} and the bias on V_{rp} . (c) Wafer-level measurement setup, where an Arduino is used to manage the logic circuitry for programming the RRAMs, and a B1500 Device Parameter Analyzer is used to read the conductance of the devices. (d) Nine cumulative distributions of the RRAM conductance, G , resulting from iterative programming of each device in an array of 4096 Resistive Random Access Memory (RRAM) devices over a range of SET programming currents. (e) The voltage developed on the V_{syn} node from the synaptic dynamic block. V_{syn} starts from the initial value close to the power supply, 1.2 V. The input spike arrives at time 0, and the voltage decreases as the capacitor M_9 discharges. The amount of discharge depends on the I_w current which is a function of the conductance value of the RRAM, G . We have recorded the V_{syn} curve for 4 conductance values. (f) Due to an input pulse train (gray pulses) at $V_{in} < 0 >$ the V_{mem} of the zeroth neuron integrates an increasing amount of voltage (purple trace) until, after six pulses, the neuron fires (light blue trace). As a result of the feedback connection to the other neuron column, neuron 1 also exhibits an increase in its membrane voltage. (g) Statistical measurements on the peak of the membrane potential in response to a pulse over an array of 5 neurons, measured across 10 cycles. (h) The output frequency of the neuron is a linear function of the conductance of the RRAM, G . The error bar reflects the variability across 4096 devices.

Figure 5. Experimental measurements of the fabricated routing tile circuits. (a) The routing tile, a crossbar whose memory state steers the input spikes from different directions towards the destination. (b) Detailed schematic of one row of the fabricated routing circuits. On the arrival of a spike to any of the input ports of the routing tile, $V_{in} < i >$, a current proportional to G_i flows in i_{in} , similar to the neuron tile. A current comparator compares this current against a reference current, I_{ref} , which is a bias current generated on chip. If $i_{in} > i_{ref}$, the spike will get regenerated, thus “pass”, or is “blocked” otherwise. (c) Wafer-level measurements of the test circuits through the probe station test setup. (d) Measurements from 4 kb array shows the Cumulative Distribution Function (CDF) of the RRAM in its High Conductive State (HCS) and Low Conductive State (LCS). The line between the distributions that separates the two is considered as the “Threshold” conductance, which the decision boundary for passing or blocking the spikes. Based on this Threshold value, the I_{ref} bias in panel (b) is determined. (e) Experimental results from the routing tile, with continuous and dashed blue traces showing the waveforms applied to the <N> and <S> inputs, while the orange trace shows the response of the output towards the <E> port. The <E> output port follows the <N> input, resulting from the device programmed into the HCS, while the input from the <S> port gets blocked as the corresponding RRAM device is in its LCS. (f) A binary checker-board pattern is programmed into the routing array, to show a ratio of 10 between the High Resistive and Low Resistive state, which sets a perfect boundary for a binary decision required for the routing tile.

7. I understand that Table 1 might be purely simulations or simple calculations. If so, just state it. However, I am not sure of what size is being compared in tasks A and B. Specifically, how are you accounting for all the communication, cooling, and other overheads that the other chips have to deal with? Are tasks A and B the size of an entire chip in the other cases? In other words, it would be unfair to compare a small problem being solved on your chip to the same small problem being solved on a GPU if the problem occupies only 1% of the GPU's circuits. At the least state this size, and be upfront about the limitations of your analysis (and its scale). It would also help to have an additional row in Table 1 showing the scale of each chip. Finally, please include full details of how you arrived at your benchmark numbers for all the chips for each task, so that any reader can easily reproduce your analysis.

Reply 8

We thank the Reviewer for analyzing Table 1 and bringing up his point, which is fair. We agree that the scale of the different chips influences energy efficiency, especially when dealing with small-scale computation, leading to the under-utilization of computing resources. The solution we propose and that we use to compile the table, is to only consider the reported energy for routing one spike, what is known as the energy “per-hop”. We base these values on what has been reported in the chips' respective papers¹⁻⁵ and cross-validated with this comprehensive review on neuromorphic processors⁶.

The routing energy estimate for the different tasks is then simply obtained by multiplying the routing energy per spike, by the total number of spikes required for solving the task. The simulations suggest that thanks to the Mosaic-aware sparsity loss terms, 95% of the spikes are routed with 0-hops operation (within a Neuron Tile), while around 4% of the spikes require a 1-hop operation (going through a Routing Tile) and the remaining necessitate k-hops (through multiple Routing Tiles). For Mosaic, the Routing energy consumption is estimated accounting for the total spike count and the routing hop partition mentioned above. This method makes the comparison more fair between Mosaic and the mentioned neuromorphic processors as it avoids accounting for all the necessary overheads of large-scale chips (communication, cooling, and others), focusing on the energy required to route spiking events only.

However, it is true that ultimately, the energy per hop is a function of how much connectivity is allowed between the computing cores. The higher the number of allowed connections, the higher the required memory to store the connectivity, and thus the larger the energy that is required to read from this memory. Each neuromorphic platform is optimized for a different objective, and this is what makes the comparison between the neuromorphic platforms difficult. Indeed, in a recent white paper from the community, we did mention this as a challenge of benchmarking neuromorphic systems⁷. But the message we want to give here is that although Mosaic is specifically optimized for minimizing the energy for communications between the different cores, it can still perform very well on the edge-size applications. This means that indeed, at least for the size of the edge problems, there is no need for large connectivity between the cores, which is what we exploit for reducing the energy.

[Action] We clarified the procedure to estimate the routing energy in the main text 211-216 of the diff file. We have also clarified further in Table 1, the source from which we have gotten the reported 0-hop, and 1-hop energy numbers. We have added a paragraph to discuss the points mentioned above regarding the trade-offs, in lines 224-228 of the diff file. The added text is copied here for your convenience.

Neuromorphic platforms have each been designed to optimize for different objectives, and the fact that the Mosaic is so much more efficient in the communication energy is that it is explicitly optimized for this thanks to its small-world connectivity layout. Despite this, as was shown in Fig.4, the Mosaic does not suffer from a considerable drop in accuracy, at least for problem sizes of the sensory processing applications on the edge. This implies that for these problems, large connectivity between the cores is not required, and thus this can be exploited for reducing the energy.

Response to Reviewer 2

This engaging paper thoroughly presents the implementation of a neuromorphic processor using RRAM (Resistive Random-Access Memory). Capitalizing on the small-world property, it effectively achieves tight coupling of neurons within a neural tile via internal connections, while maintaining loose coupling between neural tiles through RRAM-based routers. I have several questions about this paper and would appreciate the opportunity to discuss them with the authors.

Reply 9 We thank the reviewer for their comments and are humbled that they found it engaging. Please see the detailed response to your questions below.

1) The explanation of the routing tile mechanism in the paper was not entirely clear to me. What does the routed message entail? How is the routing algorithm determined and integrated into the router implemented in this paper?

Reply 10 We are sorry that this was unclear. The routed message entail a spike. The neurons in the neuron tile integrate information from their neighboring tiles, and once the value of this integration is higher than a threshold, they emit a spike. This spike is sent out to 4 directions, North (N), East (E), West (W) and South (S). The N, E, W, S routers receive these spikes as voltage pulses. If the conductance of the corresponding RRAM in the router is high/low, the input is passed onwards/blocked. The connectivity matrix of the trained small-world network, which is done offline, is mapped onto the Mosaic. The training is done in a Mosaic-aware fashion, taking into account the architecture, e.g., number of neurons per neuron tile, number of possible connections in the routing tile, and penalizing non-local connections.

[Action] To clarify how the routing is done, we have added a sentence, explaining the nature of the routing information in line 122-123.

Thus, the routed message entails a spike, which is either blocked by the router, if the corresponding RRAM is in its High-Resistive State (HRS), or is passed otherwise.

Moreover, in the section “Application to real-time sensory-motor processing through hardware-aware simulations”, we have made clarifications about what hardware awareness entails:

Therefore, case (iii) is fully hardware-aware, including the architecture choices (e.g., number of neurons per neuron tile), connectivity constraints, noise and quantization of weights.

The Mosaic aware training ensures the mappability of the trained connectivity matrix into the Mosaic.

For training case (ii), we introduce a Mosaic-regularized cost function during the training, which leads to a learned weight matrix with small-world connectivity that is mappable onto the Mosaic (see Methods for details).

2) Concerning the neuron tile, it appears to significantly differ from existing digital circuit-based systolic circuits. While input spikes can propagate to other neurons, this functionality seems achievable with conventional crossbar-based RRAM as well. Therefore, further clarification is needed to differentiate these two approaches.

Reply 11 Yes, this is correct. The neuron tile working principle is exactly like conventional crossbar arrays, implementing matrix-vector-multiplication and neuronal non-linearities. Single Neuron Tile corresponds to a single processing element (PE) in systolic arrays where partial computations occur. As data flows from PEs to PEs in a rhythmic fashion in systolic arrays, neural signals flow from Neuron Tiles to Neuron Tiles through Router Tiles in our Mosaic architecture. Hence, the basic principle of minimizing costly long-range communication energy, modular and easier configuration and 2D directional dataflow is shared between the Mosaic and systolic arrays⁸.

[Action] To clarify the details of the architecture, we have completely re-drawn Fig. S2 (now Fig. S4). This figure now provides a lot more information about the details of each crossbar, and how information flows in/out of neuron and routing tiles. We have also provided a more detailed explanation in the caption of S4 (copied in Fig. 2 of this document), and Supplementary Note 4. We hope that it is now more clear how the neuron tiles basically work like a conventional in-memory computing RRAM array with spiking neurons, and how information flows in the systolic array based on the state of resistive memory in the routing tiles.

3) How is the reference current, which determines whether an input spike passes through the router, established and provided? Who is responsible for this?

Reply 12 Thank you for the question. I_{ref} is a bias current that is generated through providing a DC voltage to the gate of a transistor on the chip. In our design, this voltage is given as an input voltage from an I/O pad. We do realize that indeed the caption of Fig. 3 was not clear about the generation of I_{ref} .

[Action] In the caption of Fig. 3 and also in the main text lines 128-130, we have clarified how I_{ref} is generated. The modified text is copied below for your convenience.

A current comparator, highlighted in the box, compares i_{buf} against i_{ref} , which is a bias generated on chip by providing a voltage from the I/O pad to the gate of a transistor (not shown in the Fig.). The I_{ref} value is decided based on the “Threshold” conductance boundary in Fig. 3d.

4) How are connections between the router tile and neuron tile formed? What characterizes the communication between them? (i.e., what constitutes the message?)

Reply 13 Dear Reviewer, thanks for bringing up this point, and allowing us to clarify the Mosaic architecture further. The output of the neuron/routing tile is hardwired to its neighboring routing/neuron tile. The communication is through a single voltage pulse, spike. The incoming spikes to the neuron tile are integrated by the neuron in its membrane potential, and once it passes its threshold, it generates a spike. This output is a wire directly connecting to the neighboring router inputs. The spike will pass through the memristor in the routing tile, generating a current. If the conductance of the memristor is high, it will be a high current, and if the conductance is low, it will be a low current. This current is compared against a reference current I_{ref} by the current comparator. The current comparator output goes high if the current from the memristor is higher than I_{ref} , and it will stay low if otherwise. This means that a spike that goes through a high conductance memristor will "pass", and the otherwise gets blocked. This is precisely the strength of this architecture which makes the connectivity easy through wiring, without suffering from long wires, as the maximum length of a wire is the size of the wire from one row/column, plus the size of the connecting column/row.

[Action] We have clarified the connection between the router and neuron tile in the supplementary Fig. S4 which is shown in Fig. 2 in this document. In this Fig. it is clarified that for instance $V_{out} < 3 : 0 >$ is the same as the $V_{in,W}$, and $V_{in,E} < 3 : 0 >$ is the same as $V_{out,W}$. We have also written in the Supplementary Note 4, that this simple wiring between the neighbours is a strong point of the Mosaic architecture.

[...]. This highlights the strength of this architecture which makes the connectivity easy through simple wiring to the neighbour, without suffering from long wires, as the maximum length of a wire is the size of the wire from one row/column, plus the size of the connecting column/row.

5) What technique was employed to measure the power consumption of the entire chip?

Reply 14 Dear Reviewer, in order to estimate the power consumption of the chip we utilized system-level SPICE simulations. In the design phase, we made great efforts to improve the energy efficiency of the circuits, while preserving their performance and - most of all - their yield. We found out that a critical parameter that influences the chip's efficiency is the pulse width of the spiking events. Spikes are voltage pulses that are applied to the 1T1R devices to read their conductance and produce the reading current, which is later integrated by the DPI synapses and LIF neurons. Ideally, a shorter pulse width for the spikes assures great energy efficiency. However, too short a pulse width results in lower precision in reading the RRAMs and - due to the effect of parasitics - might even not be propagated correctly. Thanks to extensive Monte-Carlo simulations, we made sure to find the best compromise for the pulse width so that 99.7% (3σ) of the spikes are correctly propagated. For the record, this pulse width corresponds to around 100 ns.

[Action] We added a comment on the importance of the spike's pulse width and its role in energy efficiency at the system level. We also reported how we optimized the pulse width to ensure the correct propagation of the spikes across the chip, as well as how we estimated the chip's power consumption in lines 293-299 of the diff file. This copied here for your convenience.

A critical parameter in both Neuron and Routing Tiles is the spike's pulse width. Minimizing the width of spikes assures maximal energy efficiency, but that comes at a cost. If the duration of the voltage pulse is too low, the readout current from the 1T1R will be imprecise, and parasitic effects due to the metal lines in the array might even impede the correct propagation of either the voltage pulse or the readout current. For this reason, we thoroughly investigated the minimal pulse-width that allows spikes and readout currents to be reliably propagated, at a probability of 99.7% (3σ). Extensive Monte Carlo simulation resulted in a spike pulse width of around 100 ns. Based on these SPICE simulations, we also estimated the energy consumption of Mosaic for the different tasks presented in Figure 4.

6) When making comparisons, how were the results for mosaic SNN FP32 obtained? Since RRAM cannot achieve high-precision FP32, if it was implemented using digital circuits, why is there a discrepancy between the results compared to software?

Reply 15 Dear Reviewer, we are happy to clarify this point. While it is true that RRAMs suffer from variability and hence are limited-precision devices, the iterative programming procedure⁹ mentioned in the main text allows exploiting the analog nature of the devices. As one can set a tolerance in the iterative programming procedure, there is a compromise between the number of programming iterations and the precision of the conductance state. The training procedure we utilized involves learning offline (in a conventional GPU) and then transferring the trained weights onto the crossbar array. The offline trained weights are FP32, and that is where the results for the FP32 mosaic originate. Concerning the transfer to the RRAMs, we set a tolerance of 5% of error in the conductance state. Such error inevitably determines a small accuracy drop, which is however minimized thanks to a specific training procedure that is described in both the Method section and here¹⁰ in more detail. This procedure injects noise during the training offline phase making the network more robust to noise during inference. While this accuracy difference is

minimized, the loss of precision in the weights inevitably results in a minimal performance drop in the RRAM-based Mosaic.

[Action] To make the hardware-aware offline training procedure clearer, we have modified the "RRAM-aware noise resilient training" paragraph in the Method section, in lines 359-364 of the diff file. This is copied here for your convenience:

The strategy of choice for endowing Mosaic with the ability to solve real-world tasks is offline training. This procedure consists of producing an abstraction of the Mosaic architecture on a server computer, formalized as a Spiking Neural Network that is trained to solve a particular task. When the parameters of Mosaic are optimized, in a digital floating-point-32-bits (FP32) representation, they are to be transferred to the physical Mosaic chip. However, the parameters in Mosaic are constituted by RRAM devices, which are not as precise as the FP32 counterparts. Furthermore, RRAMs suffer from other types of non-idealities such as programming stochasticity, temporal conductance relaxation, and read noise^{9,11-13}.

7) It would be helpful to provide specific data concerning the loss of precision for the three benchmarks.

Reply 16

Thank you for pointing this out. Indeed, having a specific data would help the reader to have a precise measure of the loss.

[Action] In Fig. 4, we have included the median value for all the benchmarks on top of the box of the box plots.

8) Does this chip include any peripheral circuitry? How are input and output implemented?

Reply 17 Dear Reviewer, this question allows us to clarify how the chip's periphery is organized. As the I/O of the chip is formed by simple voltage pulses (spikes), the communication from and to the chip is relatively straightforward. We simply applied voltage buffers to ensure the correct transmission of the spike from and to the chip, via probe pads. Concerning the access to the RRAM's memory array, each line in the array (Word, Bit, and Source) is accessed via a Multiplexer (MUX) that connects to probe pads. Simple logic circuits allow to control the MUXes to be able to read and program each device individually, as well as read multiple devices in parallel by row.

[Action] To make the above comment clear in the main text, we updated the Method's subsection "Fabrication/Integration" (line 309-310 of the diff file). The text is copied here for your convenience:

Multiplexers operated by simple logic circuits enable to select either a single device or a row of devices for programming or reading operations. The circuits integrated into the wafer, were accessed by a probe card which connected to the pads of the dimension of $[50 \times 90] \mu m^2$.

In conclusion, I find this paper quite intriguing and look forward to further discussions with the authors.

Reply 18 We are very glad you found it intriguing. :) We thank you for the very helpful comments, which definitely helped clarifying the ambiguities in the paper.

Response to Reviewer 3

This article by Dalgaty et al. reports on a new neuromorphic architecture, "Mosaic", aimed at implementing small-world spiking neural networks efficiently. The architecture mainly consists of neuron tiles interspersed among routing tiles, allowing for dense intra-tile connectivity and sparse inter-tile connectivity. These tiles are designed using HfO₂ RRAM as either synaptic or routing matrices, alongside post-RRAM CMOS circuitries in the 130 nm technology. Three different tasks were simulated, achieving competitive performance figures. The work presents a novel method of implementing SNNs of varying scales on hardware, which is an important topic. However, several comments and concerns need to be addressed.

Reply 19 We thank the reviewer for acknowledging the novelty of our work and for asking great and to-the-point questions which was very helpful in improving the work. Please see below the detailed response to your comments and concerns.

1. It is not entirely clear what the purpose of Figure 2c is. This figure shows the CDF of conductance states programmed using a single SET pulse, but the RRAMs are programmed using a write-and-verify strategy when used in the task demonstrations. It does not reflect the actual weight transfer non-idealities, nor does it show the range and number of possible states. It might also create some confusion at first glance as it shows a ratio of <3 between the highest and the lowest states, which is very low. Also, it would be good if the authors provide more device- or array-level characterizations in addition to the CDF of conductance states.

Reply 20

Thank you for pointing out the lack of clarity for Fig. 2c. The caption was indeed wrong. The CDF was collected through the program-and-verify and not a single pulse. The CDF shows the available range of conductances in both the low conductive and high conductive states. Indeed, we are able to program the devices to any values in this range with a margin of 5%⁹. The ratio of < 3 is only in the high conductive state, and the overall Ron/Roff ration is about 10. Other than the device level measurements, we also included some array-level level measurements. Please see the actions below based on your suggestion.

[Action] We have corrected the caption of Fig. 2c, to indicate that these distributions are indeed obtained through program-and-verify. Fig. 2 has been changed to make it clearer. We have added to Fig. 2e (previously 2c) the Low Conductance State (LCS) which now shows the full range of resistance from low resistive (LCS) to high resistive state (HCS). As you can see the on/off ratio of the full range (HCS/LCS) is >10. The ratio of 3 that was shown before was only in HCS. Moreover, we have performed new array-level measurements, showing the membrane potential statistics of neurons in an array, as a result of programming the RRAMs at a range of conductance levels. This is now shown in the added subfigure Fig. 2g.

Fig.3 has also been changed for more clarity. Also, we have performed array-level measurements for the routing tile, with RRAMs programmed to a checker board binary pattern, exploiting the full range of resistivity and providing a clear binary decision boundary for passing or blocking of the spikes between the tiles.

For your convenience, Fig. 2 and 3 of the main text are copied in this document in Fig. 4 and Fig. 5 respectively.

2. It seems that the synaptic arrays only accept positive weights. The synaptic part of the neuron circuit does not support commonly used strategies such as differential conductance to realize negative weights. Can the authors clarify how should negative weights be mapped onto the proposed hardware?

Reply 21

Indeed, in our fabricated arrays, we have not included the mechanism for negative weights. We have previously worked on circuits that consider negative weights through a differential architecture¹⁴. In this architecture, the current from the positive weight is pushed to the membrane capacitance of the neuron, thereby having a positive effect, and the current from the negative weight is pulled from it, thereby having a negative effect. However, for this design and tape-out, we were mainly focused on the concept and functionality of the array, not the final version, and we did not include the negative weight mechanism in the fabricated chips. Of course this will be taken into account in our future work.

[Action] we have clarified this in the text, lines 116-118 of the diff file, and cited our previous work that suggests how to include the negative weights in the differential architecture. This sentence is copied her for your convenience.

It is worth noting that this implementation does not take into account the negative weight, as the focus of the design has been on the concept. Negative weights could be implemented using a differential signaling approach, by using two RRAMs per synapse¹⁴.

3. Logically, when each neuron fires, the inter-tile spike outputs towards the 4 directions should be identical copies of the neuron output itself. Can the authors clarify this and make it clear in the main text? Also, Figure S2 gives the impression that the inter-tile outputs towards each of the 4 directions originate from different neurons, which creates confusion if the aforementioned case is true.

Reply 22 Thank you again for noticing this lack of clarify. Indeed, the output of the neurons from neuron tiles are copied in 4 directions, and the conductance state of the RRAM in each core decides if the spike is propagated and to which direction.

[Action] We have completely redrawn Fig. S2 (now S4) which is shown in Fig. 2 in this document. In this Fig., it is now clarified that the output of the neurons are copied in 4 directions and that the input to the neuron/routing tile is from the neighboring routing/neuron tile. The details of the connections are also illustrated, for instance $V_{out} < 3 : 0 >$ is the same as the $V_{in,W}$, and $V_{in,E} < 3 : 0 >$ is the same as $V_{out,W}$. We have also written in the Supplementary Note 4, that this simple wiring

between the neighbours is a strong point of the Mosaic architecture. We added two sentences in lines 65 and 67 to clarify how the signaling is done between the tiles. We copy them here for your convenience.

The spikes from the neuron tile are copied in four directions of N, S, E and W. These spikes are communicated between neuron tiles through a mesh of blue routing tiles, whose crossbar array stores the connectivity pattern between neuron tiles. The routing tiles at different directions decided whether or not the received spike should be further communicated.

4. Is the size of each neuron tile and routing tile fixed during the fabrication of the entire hardware? If so, is there an optimal size for these tiles such that networks with a wide range of sizes can be efficiently implemented?

Reply 23 Dear Reviewer, thank you for this interesting question. Yes, the size of Neuron and Routing Tiles have to be fixed during the design phase. The designer can optimize for a desired memory footprint by deciding the *number of Neuron Tiles* and the *number of neurons per Neuron Tile*, which in turn determines the available neural resources for the targeted downstream task.

Regarding the optimal selection of the number of neuron tiles for reduced memory footprint, we calculated the number of required memristive devices as a function of the number of neurons per tile; for a wide range of network sizes in Fig. 1g of the main text. We showed that indeed the implementation of scaled small-world graphs is done more efficiently on the Mosaic architecture.

Our reported network configurations with the speech dataset (64 neuron tiles with 32 neurons each) and reinforcement learning (16 neuron tiles with 16 neurons each) are in this regime where Mosaic architecture provides smaller memory footprint implementation.

[Action] Regarding the optimal selection of neuron tiles for task performance, we added a new Supplementary Note 8, Fig. S10 that compares the effects of various neuron tile sizes and the total number of neurons on the SHD dataset test accuracy. This is copied in the Fig. 6 of this document for your convenience.

Our results suggest that the networks that are too sparse, i.e. with small Neuron Tiles generally perform badly. This is intuitive as the recurrent parts get smaller, the interaction between different spatio-temporal features gets weaker. We are planning to investigate the impacts of tile design parameters and sparsity, i.e. small-world scaling laws, on the network performance, capacity, and generalization capabilities in future works.

5. It is interesting to see that the Mosaic constrained network shows little to no performance degradation for the chosen constraining parameters. Nevertheless, did the authors experiment with further reducing the number of neurons per neuron tile to see the degree to which tighter constraints impact network performance? Could this serve as a guide to designing optimally sized tiles for the architecture?

Reply 24 We thank the reviewer for this insightful question. While benchmarking Mosaic, we treated the number of neurons in the Neuron Tile as a hyper-parameter and ran multiple experiments to find the optimal point for performance that is compatible with hardware constraints. While doing so, we found out that the performance decreases if the size of the Neuron Tile is selected to be too small. However, we also found that the total number of neurons in the network is the parameter that plays the most drastic role in the network accuracy.

Moreover, there are other hyper-parameters that should be tuned along with the Neuron Tile size. In particular, the sparsity loss terms for connections in the Mosaic affect the network's classification accuracy. To keep things simple, we have not tuned all hyper-parameters for each Neuron Tile size. Our conclusion is that while an optimum Neuron Tile size exists, this hyper-parameter is not the most crucial in determining the Mosaic architecture performance. This means that fixing Mosaic's Neuron Tiles size to a predetermined value - for example 64 neurons - is a guarantee for good enough performance across different tasks.

We are going to investigate the impacts of tile design parameters and sparsity, i.e. small-world scaling laws, on the network performance, capacity and generalization capabilities in a future work to provide with a guide for designing efficient Mosaic tiles.

[Action] Regarding the optimal selection of neuron tiles for task performance, we added a new Supplementary Note 8, Fig. S10 that benchmarks various neuron tile sizes and total number of neurons for the SHD dataset test accuracy and Supplementary Note 9, Fig. S11 that demonstrates the impact of sparsity regularization during the training (copied here as Fig.6 and Fig. 7). We clarified this in the main text as following (lines 182-185 of the diff file).

Furthermore, we experimented with various numbers of Neuron Tiles and the number of neurons within each Neuron Tile (Supplementary Note 9, Fig. S10), as well as sparsity constraints (Supplementary Note 9, Fig. S11) as hyperparameters. We

found that optimal performance can be achieved when an adequate amount of neural resources are allocated for the task.

Figure 6. SHD keyword spotting dataset test accuracies for Mosaic architectures with different total number of neurons in the network for a) 4x4 Neuron Tile layout (a total of 16 number of Neuron tiles) and b) 8x8 Neuron Tile layout. The number of neurons per tile is equal to the total number of recurrent neurons divided by the number of neuron tiles. Median and standard deviation are calculated using 3 experiments with varying sparsity constraints.

Figure 7. SHD keyword spotting dataset test accuracies for Mosaic architectures trained with different sparsity regularization values. As explained in the Methods section of the main text, the regularization is added to the loss function to exponentially penalize the long-range connections. The plot shows the accuracy for strong (default, $\lambda = 0.1$), medium ($\lambda = 0.05$) and weaker ($\lambda = 0.01$) sparsity regularization on a) 4x4 neuron tile layout and b) 8x8 neuron tile layout. Median and standard deviation are calculated using 4 experiments with varying number of neurons per neuron tile.

6. The network performance should decrease with a smaller network scale. How does this trade-off of a Mosaic-constrained network compare to that of a conventional network? If only a certain number of memory units are allocated for a network, is it possible for a conventional RSNN (with fewer neurons) and a Mosaic-constrained network (with more neurons) to achieve similar performance figures? Ruling out such a possibility will further strengthen the argument that small-world networks and, accordingly, the proposed architecture is a better way of doing neuromorphic computing.

Reply 25 Dear Reviewer, this is yet another very stimulating reflection, thank you for bringing this up. The aim of our work is to show an advantage of our proposed architecture at the hardware level, with the energy efficiency for routing events being the target figure of merit. However, the computational power of small-world networks is definitely an aspect we are very interested in. For this, we believe that more efforts should be dedicated to finding novel training procedures that better make use of sparse networks (or even small-world graphs in particular). This is an open and fascinating research question and something we are willing to invest time into in the future.

For this work, we treated Mosaic as a conventional SNN, training with plain BPTT and simply adding loss terms that punish the network's dense connectivity and shape sparse graphs. For this reason, the potential advantages of small-world architectures do not naturally emerge and the performance of the network is mainly related to the number of parameters. In fact, we found out that Mosaic requires more neurons, but about the same number of parameters, to perform at the same level of an RSNN. This confirms to us that taking advantage of small-world connectivity requires a novel training procedure, which we hope to

develop in the future. Moreover, here, we are benchmarking our system on sensory processing tasks, and taking advantage of the small-worldness for energy savings, by keeping the information local. However, these tasks do not necessarily take advantage of the small-worldness from a computational perspective. We have to yet discover and understand the tasks that exploit the small-world connectivity from a computational standpoint.

[Action] We have added a discussion part in now the “Discussions and Conclusions” section based on the reply above. This is reflected in lines 249-263 in the diff file which is copied here for your convenience.

In this work, we have treated Mosaic as a standard Recurrent Spiking Neural Network (RSNN), and trained it with BPTT using the surrogate gradient approximation, and simply added the loss terms that punish the network’s dense connectivity to shape sparse graphs. Therefore, the potential advantages of small-world architectures from a computational perspective do not necessarily emerge, and the performance of the network is mainly related to the number of parameters. In fact, we found out that Mosaic requires more neurons, but about the same number of parameters, to perform at the same level of an RSNN. This confirms to us that taking advantage of small-world connectivity requires a novel training procedure, which we hope to develop in the future. Moreover, here, we are benchmarking our system on sensory processing tasks, and taking advantage of the small-worldness for energy savings, by keeping the information local. However, these tasks do not necessarily take advantage of the small-worldness from a computational perspective. In future works, one can foresee tasks that can exploit the small-world connectivity from a computational standpoint.

7. The workings of the Mosaic constrained network, in particular the input stage of the network, are not clear. Are all the elements of the input vector linearly transformed first before being accumulated by the neurons, or is each input element only accumulated by their corresponding neuron? Figure S8 seems to support the former, but the description in the Methods section seems to support the latter. The concern here is that if the inputs span multiple tiles as in the case of the KWS and RL tasks, and there is an initial linear transformation, inputs into one tile cannot contribute (or can only weakly contribute) to the neuron state of other input-receiving tiles. The input channels are thus automatically grouped based on the tiles they are situated in and might contribute unequally to the network. In other words, if a set of input channels contains more useful information than others, then there might exist a way of grouping these input channels to maximize network performance. Also, for the KWS task, the neurons in the input tiles are not recurrently connected. Is it the same for the other tasks? Similarly, if these input neurons span multiple tiles and are still recurrently connected, they might also contribute unequally to the network.

Reply 26

While it is true that in many architectures that use convolutional and fully-connected layers all of the weights globally act on all of the inputs, other more recent approaches to deep learning process input data in a fashion similar to how we have proposed to do so with the Mosaic. In particular Vision Transformers with local attention¹⁵ and some so-called MLP-mixer architectures¹⁶ operate by repeatedly transforming local data “patches” using local information only. In some tasks, this type of architecture has in fact been found to be favorable to global transforms. It is our hope that small-world architectures based on the same principles can be compiled into the Mosaic and that this in fact will not pose a problem. Furthermore, we remark that in biological nervous systems many sensory modes, in particular, the visual system like that of the fruit fly¹⁷, input data is also treated in a localized “patch-like” fashion and this could be a good guide in the development of bio-inspired computing hardware.

[Action] We added a brief explanation in the Discussion phase on why we believe that the local computation in Mosaic might not be a constraint, but rather a feature to exploit (Lines 264-268 of the diff file, copied here for your convenience).

Mosaic favors local processing of input data, in contrast with conventional deep learning algorithms such as Convolutional and Recurrent Neural Networks. However, novel approaches in deep learning, e.g., Vision Transformers with Local Attention¹⁵ and MLP-mixers¹⁶, treat input data in a similar way as Mosaic, subdividing the input dimensions, and processing the resulting patches locally. This is also similar to how most biological system processes information in a local fashion, such as the visual system of fruit flies¹⁷.

8. It would be good if the authors provide more detail on how the energy/power figures of the architecture were obtained/calculated. Besides, the proposed architecture requires more routing tiles than neuron tiles, and a routing tile has 4x the number of post-array circuitries (op-amp, current comparator) than a neuron tile, all of which will add up to the total power consumption. How significant is this, especially when compared with using a simple crossbar array architecture? This is important as the deployment of the proposed hardware is aimed at extreme-edge applications, as mentioned by the authors.

Reply 27

That is a great point. Indeed, although the connectivity is sparse, we still require more number of routing nodes than computing nodes. However, the routing tiles are much more compact, since they only require a binary classification. This means that the read-out circuit does not require as large of an SNR compared to the neuron tiles, since the neuron tiles need to distinguish between more resistances. That reduces the overhead of the routing tiles readout in terms of both area and power. Despite that the layout of the circuit was not optimized for density, since the fabricated chip was meant as a demonstrator, the Footprint area of the Routing Column [44x6.56] μm is around 33% that of the Neuron Column [131.4x6.56] μm . In fact, most of the space in the Routing Column is taken by the front-end circuitry that reads out current from the RRAM device. We envision an optimized version of the Routing Column in which a dedicated and simplified readout circuit reduces its complexity and footprint area. [Action] We have added a point about this in the discussion which is copied here for your convenience. (lines 249-252 of the diff file).

Although the connectivity of the Mosaic is sparse, it still requires more number of routing nodes than computing nodes. However, the routing tiles are more compact than the neuron tiles, as they only perform binary classification. This means that the read-out circuit does not require a large Signal to Noise Ratio (SNR), compared to the neuron tiles. That reduces the overhead of the routing tiles readout in terms of both area and power.

Minor issues: 9. On page 8 line 202: "The final row of Table 1 indicates the power consumption of all of the hardware in task A (ECG anomaly detection)." The second last row and final row refer to routing power for task A and task B, respectively.

Reply 28 Dear Reviewer, thank you very much for spotting this! It is now fixed.

[Action] We updated lines 237 in the diff file.

10. On page 9 line 222: "(due to the 90 degrees phase-shift between the gate and drain of transistor M1 in Fig. 2)" The phase shift between the gate and drain of M1 should be 180°, not 90°.

Reply 29 Thank you very much for catching that! This is now fixed.

[Action] Dephasing fixed in line 278.

11. On page 9 line 230: Did the authors cite the wrong supplementary figure? If so, Figure S8 is not cited anywhere else in the main text.

Reply 30 Thank you for noticing the mistake, we indeed meant to cite Fig. 2 and 3 .

[Action] We adjusted the citation(lines 286 and 289 of the diff file).

12. On page 11:

From the schematic in Figures 1 and 2a, each neuron tile has dedicated intra-tile recurrent connections on top of inter-tile inputs from the 4 directions. So, is the total number of RRAM devices in each neuron tile $5k^2$ instead of $4k^2$ (k inputs from 4 directions contribute $4k^2$, recurrent connections contribute k^2)?

Reply 31 Dear Reviewer, we appreciate the meticulous attention in spotting this issue, which is in fact a mistake. In fact, we omitted the recurrent connections in the Neuron Tiles, adding up to k^2 . Overall, the correct Memory Footprint of the Neuron Tiles is $5 \times k^2$.

[Action] We adjusted the mathematical expression in lines 361-362 and equation (2). We also updated Figure 1g with the new equation.

13. On page 11 line 365: "We consider R to be the number of routing tiles with $4k^2$ devices in each." $4k^2$ should be $(4k)^2$?

Reply 32 Dear Reviewer, thank you for pointing this out.

As you correctly noticed, the number of devices per routing tile is $(4k)^2$.

[Action] We adjusted the mathematical expression in line 434.

References

1. Merolla, P., Arthur, J., Alvarez, R., Bussat, J.-M. & Boahen, K. A multicast tree router for multichip neuromorphic systems. *Circuits Syst. I: Regul. Pap. IEEE Transactions on* **61**, 820–833, DOI: [10.1109/TCSI.2013.2284184](https://doi.org/10.1109/TCSI.2013.2284184) (2014).
2. Painkras, E. *et al.* SpiNNaker: A 1-W 18-core system-on-chip for massively-parallel neural network simulation. *IEEE J. Solid-State Circuits* **48**, 1943–1953, DOI: [10.1109/JSSC.2013.2259038](https://doi.org/10.1109/JSSC.2013.2259038) (2013).
3. Benjamin, B. V. *et al.* Neurogrid: A mixed-analog-digital multichip system for large-scale neural simulations. *Proc. IEEE* **102**, 699–716 (2014).
4. Moradi, S., Qiao, N., Stefanini, F. & Indiveri, G. A scalable multicore architecture with heterogeneous memory structures for dynamic neuromorphic asynchronous processors (DYNAPs). *Biomed. Circuits Syst. IEEE Transactions on* **12**, 106–122, DOI: [10.1109/TBCAS.2017.2759700](https://doi.org/10.1109/TBCAS.2017.2759700) (2018).
5. Davies, M. *et al.* Loihi: A neuromorphic many-core processor with on-chip learning. *IEEE Micro* **38**, 82–99 (2018).
6. Basu, A., Deng, L., Frenkel, C. & Zhang, X. Spiking neural network integrated circuits: A review of trends and future directions. In *2022 IEEE Custom Integrated Circuits Conference (CICC)*, 1–8 (IEEE, 2022).
7. Yik, J. *et al.* Neurobench: Advancing neuromorphic computing through collaborative, fair and representative benchmarking. *arXiv preprint arXiv:2304.04640* (2023).
8. Kung. Why systolic architectures? *Comput. (Long Beach Calif.)* **15**, 37–46 (1982).
9. Esmanhotto, E. *et al.* High-density 3D monolithically integrated multiple 1T1R multi-level-cell for neural networks. In *2020 IEEE International Electron Devices Meeting (IEDM)*, 36–5 (IEEE, 2020).
10. Moro, F. Memristor-aware-training for resilient neural networks (2023).
11. Dalgaty, T. *et al.* In situ learning using intrinsic memristor variability via Markov chain Monte Carlo sampling. *Nat. Electron.* **4**, 151–161 (2021).
12. Zhao, M. *et al.* Investigation of statistical retention of filamentary analog rram for neuromorphic computing. In *2017 IEEE International Electron Devices Meeting (IEDM)*, 39.4.1–39.4.4, DOI: [10.1109/IEDM.2017.8268522](https://doi.org/10.1109/IEDM.2017.8268522) (2017).
13. Moro, F. *et al.* Hardware calibrated learning to compensate heterogeneity in analog rram-based spiking neural networks. *IEEE Int. Symp. Circuits Syst.* (2022).
14. Payvand, M., Nair, M. V., Müller, L. K. & Indiveri, G. A neuromorphic systems approach to in-memory computing with non-ideal memristive devices: From mitigation to exploitation. *Faraday Discuss.* **213**, 487–510 (2019).
15. Pan, X., Ye, T., Xia, Z., Song, S. & Huang, G. Slide-transformer: Hierarchical vision transformer with local self-attention. In *Proceedings of the IEEE/CVF Conference on Computer Vision and Pattern Recognition*, 2082–2091 (2023).
16. Yu, T., Li, X., Cai, Y., Sun, M. & Li, P. S2-mlp: Spatial-shift mlp architecture for vision. In *Proceedings of the IEEE/CVF winter conference on applications of computer vision*, 297–306 (2022).
17. Strother, J. A., Nern, A. & Reiser, M. B. Direct observation of on and off pathways in the drosophila visual system. *Curr. Biol.* **24**, 976–983 (2014).

REVIEWER COMMENTS

Reviewer #1 (Remarks to the Author):

The revised manuscript is good with me, and I think is suitable for this journal. Well done.

Suhas Kumar

Reviewer #3 (Remarks to the Author):

The authors have addressed most of my concerns and the paper is much improved. The paper can be published if the last few minor issues are resolved.

1. The authors still did not provide straight to the point discussion regarding the significance of the power consumption contributed by the large number of read-out circuitries in the routing tiles. Each added comparator/op-amp should draw a considerable amount of static power. It would be good to quantify/estimate how much does the relaxed SNR requirement of the routing tiles reduce the power consumption relative to the neuron tiles.
2. A minor detail: The reinforcement learning demonstration with 16 neuron tiles and 16 neurons per tile requires 155648 devices, which is in fact more than $256^2 = 65536$ devices in a fully-connected array.
3. Also, the authors should recheck all mathematical expressions in detail. E.g., in Page 12, it should be $T \times 5 \times k^2$.

Response to the reviewers:

Mosaic: in-memory computing and routing for small-world spike-based neuromorphic systems

We sincerely thank the reviewer for further spending their time on providing critical feedback to our paper. The comments are greatly appreciated and we have done our best to address them. We have addressed the changes in the main and supplementary texts, highlighted with a strike-through red when text was removed, and in blue when text was added. In this letter, we reply with black, and whenever the changed text is copied here, we highlight it in green for your attention. Thank you very much.

Response to Reviewer 3

The authors have addressed most of my concerns and the paper is much improved. The paper can be published if the last few minor issues are resolved.

Reply 1 We are glad to hear we were able to address most of your feedback. Please see below point by point reply and action to your comments.

1. The authors still did not provide straight to the point discussion regarding the significance of the power consumption contributed by the large number of read-out circuitries in the routing tiles. Each added comparator/op-amp should draw a considerable amount of static power. It would be good to quantify/estimate how much does the relaxed SNR requirement of the routing tiles reduce the power consumption relative to the neuron tiles.

Reply 2 Dear reviewer, thank you for this valuable comment. We have done an analysis on the required bandwidth of the read-out loop for the cases of neuron and routing tiles. The bandwidth of the loop determines the current and thus the power requirements in both cases. You can see below the details of these calculations.

[Action] New supplementary note 9 was added to the supplementary files. For your convenience it is copied below.

Figure 1. Analysis of the read-out circuitry. The amplifier with gain A, pins voltage V_x to the voltage V_{top} . On the arrival of the pulse on $V_{in} < i >$, a current equal to $i_{in} = (V_{top} - V_{bot})G_i$ flows into the memristor i , which is then mirrored out i_{out} .

Figure 1 details the implementation of the read-out circuit used in the Mosaic architecture. Though not optimized for area, we have used this implementation for both the neuron and routing tiles.

The dominant power consumption of the circuit depends on the required bandwidth (BW) of the feedback loop. This BW depends on the maximum conductance of the RRAM, G_{max} . For $G_{i,max}$, once an input arrives to $V_{in} < i >$, the current i_{in} has to settle to $(V_{top} - V_{bot})G_{i,max}$ within a settling time, t_s , a proportion of the pulse width. This timing sets the speed at which the loop should work, and thus its BW. If the loop does not close in this time, the amplifier will slew, and the voltage V_x drops. In both neuron and routing tiles, this condition should be met for V_x to stay pinned at V_{top} , while RRAM is being read. However, the neuron and routing tiles have different BW requirements.

In the **neuron tile**, the read-out circuitry has to resolve between at least 8 levels of current for the 8 levels that each RRAM device can take (Fig. 2d of the main text). Therefore, the Least Significant Bit (LSB) of the i_{in} current for the neuron tile is $i_{in,LSB,N} = \frac{V_{ref}(G_{max}-G_{min})}{N}$. Based on the Fig. 2d, this value for the neuron tile is $\frac{100mV(120\mu S-40\mu S)}{8} = 1\mu A$. Note that since the 8 levels to be resolved are in the Low Resistive State of the RRAM, the G_{max} and G_{min} are the minimum and maximum of the range in the LRS, which correspond to $40\mu S$ and $120\mu S$, respectively.

In the **routing tile**, the read-out circuitry has to resolve between two levels which will either let the spike regenerate and thus propagate, or will get blocked. Therefore, the LSB of the i_{in} current in the routing tile is $i_{in,LSB,R} = \frac{V_{ref}(G_{max}-G_{min})}{N}$. Based on the Fig. 2d, this value for the neuron tile is $\frac{100mV(40\mu S-10\mu S)}{2} = 15\mu A$. Note that since the 2 levels to be resolved are the LRS and HRS of the RRAM, the G_{max} and G_{min} correspond to $10\mu S$ and $40\mu S$.

To be able to distinguish between any two levels in both cases, we will consider a maximum error of $\frac{i_{in,LSB}}{2}$. Therefore, the maximum tolerable error in the the neuron tile is $0.5\mu A$ and in the routing tile is $7.5\mu A$.

This means that if the feedback loop does not close in t_s of the pulse width, V_x drop is a lot more tolerable in the routing tile than it is in the neuron tile. This suggests that the bandwidth requirements in the case of neuron tile is $7.5/0.5 = 15$ times more than that of the routing tile. The BW requirements directly translate to the biasing of the amplifier and thus its power consumption. Therefore, the static power consumption of the neuron tile is 15 times that of the routing tile. The current requirements also translate to area, since larger currents require wider transistors.

2. A minor detail: The reinforcement learning demonstration with 16 neuron tiles and 16 neurons per tile requires 155648 devices, which is in fact more than $256^2 = 65536$ devices in a fully-connected array.

Reply 3 Indeed, this observation is correct and can be verified using Fig. 1g or Eq. (1) for the Mosaic architecture described with parameters $k = 16$ and $i = 3$. In the Results section, we noted that small-sized network implementations on Mosaic might not always optimize memory footprint, as follows: "While smaller network sizes (i.e., 128 neurons) do not yield memory savings compared to a single large array, the benefit becomes increasingly significant with scaling. For instance, a network of 1024 neurons with 4 neurons per tile in the Mosaic architecture requires almost an order of magnitude fewer memory devices than a single crossbar to implement an equivalent network model." In the reinforcement learning experiments, we employed Evolution Strategies (ES) to demonstrate that Mosaic-aware training can optionally be conducted in a gradient-free manner. However, the drawback of ES is that it necessitates the parallel operation of 4096 neural networks population within a single GPU, constraining us to the use of smaller-sized networks. In addition, we expect larger networks to perform even better on given RL tasks.

[Action] We further clarified this by adding the following sentence to Reinforcement Learning task description between line 398-400.

We note that for simulation purposes, selecting a small network of 16 neuron tiles with 16 neurons each, while not optimal in terms of memory footprint (Eq.2), was preferred to fit the large ES population within the constraints of single GPU memory capacities.

3. Also, the authors should recheck all mathematical expressions in detail. E.g., in Page 12, it should be $T \times 5 \times k^2$.

Reply 4 Dear Reviewer, thank you for your comment. We revised all the equations in the paper to make sure they are now correct and we updated their format so that they read clearly. Also, for more clarity, we changed the symbol for multiplication in all the equations in Methods/Calculation of memory footprint from " \times " to the "dot" formalism (for example, $T \cdot 5k^2$).

[Action] Changes are visible in line 422 and in Equation (2).

REVIEWERS' COMMENTS

Reviewer #3 (Remarks to the Author):

The authors have made satisfactory revisions to address the remaining issues. I have no further comments and can recommend the work for publication in Nature Communications.